# Identification of fatty acid amide hydrolase as a metastasis suppressor in breast cancer

Isabel Tundidor[1,2], Marta Seijo-Vila[1,2], Sandra Blasco-Benito[1,2], María Rubert-Hernández[1,2], Sandra Adámez[1], Clara Andradas[3,4], Sara Manzano[5], Isabel Álvarez-López[5,6], Cristina Sarasqueta[7,8], María Villa-Morales[9,10], Carmen González-Lois[11], Esther Ramírez-Medina[12], Belén Almoguera[12], Antonio J. Sánchez-López[13,14], Laura Bindila[15], Sigrid Hamann[16], Norbert Arnold[16], Christoph Röcken[17], Ignacio Heras-Murillo[18], David Sancho[18], Gema Moreno-Bueno[19], María M. Caffarel[5,20], Manuel Guzmán[1,21], Cristina Sánchez[1,2] ✉ & Eduardo Pérez-Gómez[1,2] ✉

Clinical management of breast cancer (BC) metastasis remains an unmet need as it accounts for 90% of BC-associated mortality. Although the luminal subtype, which represents >70% of BC cases, is generally associated with a favorable outcome, it is susceptible to metastatic relapse as late as 15 years after treatment discontinuation. Seeking therapeutic approaches as well as screening tools to properly identify those patients with a higher risk of recurrence is therefore essential. Here, we report that the lipid-degrading enzyme fatty acid amide hydrolase (FAAH) is a predictor of long-term survival in patients with luminal BC, and that it blocks tumor progression and lung metastasis in cell and mouse models of BC. Together, our findings highlight the potential of FAAH as a biomarker with prognostic value in luminal BC and as a therapeutic target in metastatic disease.

Breast cancer (BC) is a major public health issue with more than half a million deaths worldwide each year, principally due to metastatic dissemination[1]. At the molecular level, BC is a highly heterogeneous disease that can be stratified into different subtypes with diverse tumor characteristics and clinical outcomes[2]. Among them, luminal BC represents about >70% of BC cases and, despite relatively good prognosis, long-term (>5 years) recurrence is significantly higher compared with the other subtypes[3], and can recur up to two decades after initial diagnosis with visceral metastasis[4,5]. On the other hand, the lack of targeted therapies for endocrine-resistant luminal BC is still an unresolved clinical problem. Seeking for therapies is therefore essential, as well as for screening tools able to foresee the late relapse in an accurate and precise manner.

The endocannabinoid system (ECS) is a cell communication system involved in the control of multiple biological processes and whose dysregulation has been identified in various diseases, including cancer[6–8]. This system is archetypically composed of the G protein-coupled cannabinoid receptors (CBRs) $CB_1R$ and $CB_2R$, their endogenous ligands anandamide (AEA) and 2-arachidonoylglycerol (2-AG), and the enzymes that produce and metabolize these ligands[9]. Over the past decades, most research on the ECS in cancer has focused on the widely demonstrated notion that pharmacological activation of CBRs in mouse models of cancer evokes antitumor responses[10]. However, very few studies have addressed the role of this system in the progression of cancer. Fatty acid amide hydrolase (FAAH) is the key enzyme responsible for the degradation of AEA and, as such, determines the availability and biological activity of this lipid messenger[11,12]. Here, we unveil that (i) FAAH expression in BC is highly associated with luminal BC, where patients with low tumor FAAH levels show significantly worse overall survival; (ii) its expression in BC cells promotes a shift of the tumor phenotype towards a more differentiated state; and (iii) it reduces tumor progression and metastasis in cellular and

mouse models of BC. Taken together, these findings establish the role of FAAH as a metastasis suppressor in BC and suggest that it may constitute a prognostic biomarker and a potential therapeutic target in this disease.

## Results

### FAAH expression is associated with luminal BC

To determine whether the ECS is deregulated in BC, we analyzed the expression of the most relevant genes known to regulate endocannabinoid synthesis and degradation along the major molecular subtypes of BC as determined by the PAM50 signature[13]. Transcriptomic analyses performed by the bc-GenExMiner website[14] (which contains transcriptomic data from >15,000 BC patients) revealed that, out of the genes examined, *FAAH* was the only one showing differential expression between the different intrinsic subtypes. Specifically, significantly higher mRNA levels of *FAAH* were found in luminal tumors compared to HER2-positive and basal-like tumors (Supplementary Fig. 1a). These results were again found after the analysis of additional patient datasets, including METABRIC[15] and TCGA[16], which, between them, comprise data from roughly 3500 patients (Fig. 1a, b and Supplementary Fig. 1b). Of note, the lowest *FAAH* expression in METABRIC samples was highly associated with the IC10 cluster of tumors (Supplementary Fig. 1c), a basal-like enriched subgroup with high genomic instability and extreme probability of relapse or cancer-related death at 5 years[15,17].

To establish whether the association between higher *FAAH* mRNA levels and luminal tumor phenotypes also occurs at the protein level, we analyzed FAAH protein expression in a series of 617 human breast tumor samples included in several tissue microarrays (all here referred to as TMA #1). FAAH expression was scored as 0 (no IHC staining), 1

(weak staining), 2 (moderate staining), or 3 (high staining) by a pathologist (Fig. 1c). Here, high FAAH expression was strongly associated with hormone-dependent tumors, especially with estrogen receptor-positive (ER+) tumors (Fig. 1d). High FAAH expression was also significantly associated with low histological grades (i.e., highly differentiated tumors), which is consistent with the fact that most luminal tumors display grade 1 (G1)/G2 features (Fig. 1d and Supplementary Fig. 1d). In addition, an inverse association between FAAH expression and HER2 and triple-negative status was detected (Fig. 1d).

Interestingly, the strong association between FAAH and ER+ tumors was further observed at the mRNA level after the analysis of numerous public DNA microarray datasets (Supplementary Figs. 2a, b), thus suggesting that FAAH expression might be connected with ER signaling. Supporting this notion, FAAH expression was upregulated by estradiol and downregulated by ER silencing[18] in luminal BC cell lines (Supplementary Figs. 2c, d), which is consistent with the presence of estradiol-responsive elements in the *FAAH* promoter[19]. In addition, we found *ESR1* to be the transcription factor that binds to the *FAAH* promoter with the highest score according to published ChIP-chip and ChIP-seq data included in Harmonizome, a collection of processed datasets developed by the Ma'ayan Laboratory at Mount Sinai[20,21] (Supplementary Fig. 2e).

Together, these observations show a strong association between high FAAH expression and luminal BC.

### Low FAAH expression in breast tumors is associated with poor patient prognosis

In order to investigate the potential link between tumor FAAH levels and patient outcomes, we analyzed the clinical data available in the TMA #1. Here, patients with low tumor FAAH expression (scores 0 and

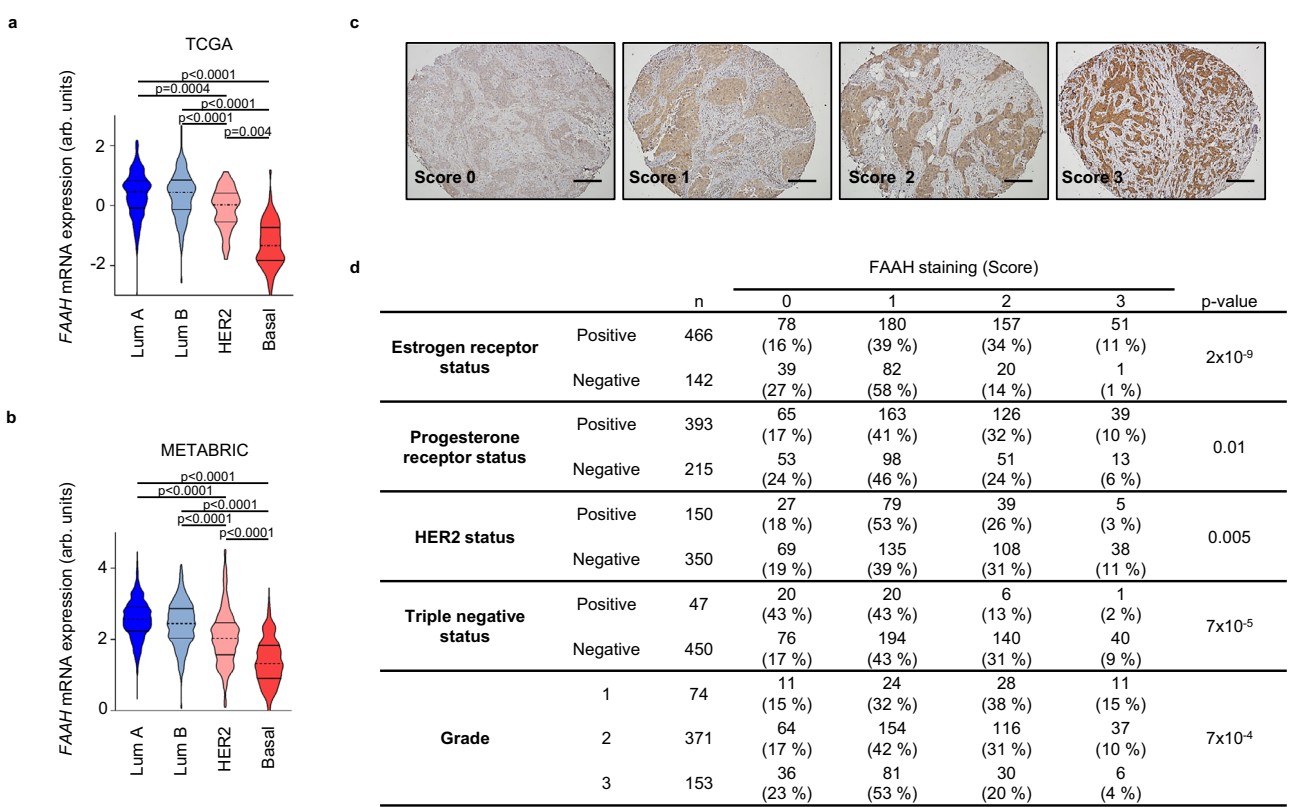

| | | n | FAAH staining (Score) | | | | p-value |
|---|---|---|---|---|---|---|---|
| | | | 0 | 1 | 2 | 3 | |
| **Estrogen receptor status** | Positive | 466 | 78 (16 %) | 180 (39 %) | 157 (34 %) | 51 (11 %) | $2 \times 10^{-9}$ |
| | Negative | 142 | 39 (27 %) | 82 (58 %) | 20 (14 %) | 1 (1 %) | |
| **Progesterone receptor status** | Positive | 393 | 65 (17 %) | 163 (41 %) | 126 (32 %) | 39 (10 %) | 0.01 |
| | Negative | 215 | 53 (24 %) | 98 (46 %) | 51 (24 %) | 13 (6 %) | |
| **HER2 status** | Positive | 150 | 27 (18 %) | 79 (53 %) | 39 (26 %) | 5 (3 %) | 0.005 |
| | Negative | 350 | 69 (19 %) | 135 (39 %) | 108 (31 %) | 38 (11 %) | |
| **Triple negative status** | Positive | 47 | 20 (43 %) | 20 (43 %) | 6 (13 %) | 1 (2 %) | $7 \times 10^{-5}$ |
| | Negative | 450 | 76 (17 %) | 194 (43 %) | 140 (31 %) | 40 (9 %) | |
| **Grade** | 1 | 74 | 11 (15 %) | 24 (32 %) | 28 (38 %) | 11 (15 %) | $7 \times 10^{-4}$ |
| | 2 | 371 | 64 (17 %) | 154 (42 %) | 116 (31 %) | 37 (10 %) | |
| | 3 | 153 | 36 (23 %) | 81 (53 %) | 30 (20 %) | 6 (4 %) | |

**Fig. 1 | FAAH is highly expressed in differentiated breast tumors.** Relative *FAAH* mRNA expression along the four molecular subtypes of BC according to the TCGA[16] (**a**) and METABRIC[15] (**b**) datasets. Data were analyzed by 1-way ANOVA with post Tukey's multiple comparison test. **c** Representative images (*n* > 10 biological replicates) showing FAAH expression scoring according to intensity staining in samples from TMA #1. Scores 0, 1, 2, and 3 correspond to no, weak, moderate, and high staining, respectively. Scale bar = 500 μm. **d** Association between FAAH expression and the molecular features of breast tumor samples included in TMA #1. Pearson's chi-squared test was used for statistical analysis. Source data are provided as a Source Data file.

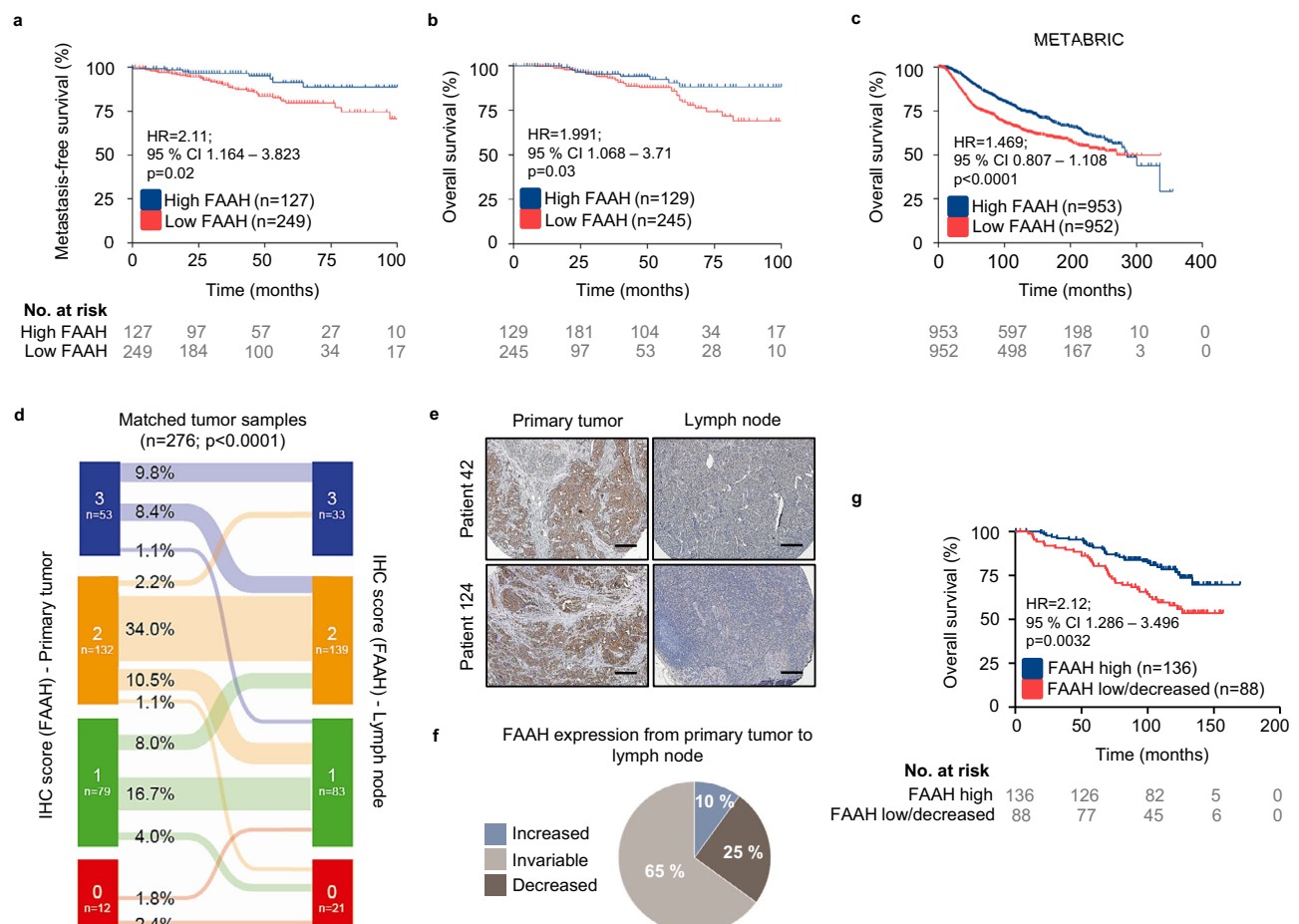

**Fig. 2 | Low FAAH expression in breast tumors is associated with poor patient prognosis.** Kaplan-Meier curves for metastasis-free survival (**a**) and overall survival (**b**, **c**) in BC samples with high and low FAAH expression obtained from TMA #1 (**a**, **b**) and the METABRIC database[15] (**c**). Kaplan-Meier curves were statistically compared by the log-rank test. **d** Matched FAAH expression in samples of primary tumor and lymph node (LN) metastasis obtained from TMA #2. Data were analyzed by 2-tailed Student's t-test. **e** Representative IHC images (*n* > 10 biological

replicates) showing FAAH expression decrease in LN metastasis compared to primary tumor. Scale bar = 250 μm. **f** Pie chart representing variations in FAAH expression from primary tumor to LN metastasis in TMA #2. **g** Kaplan-Meier curves for overall survival from patients included in TMA #2 according to the variations of FAAH expression from the primary tumor to the LN. Kaplan-Meier curves were statistically compared by the log-rank test. Source data are provided as a Source Data file.

1) had a higher probability for developing distant metastases (Fig. 2a) and a significantly decreased overall survival (Fig. 2b) than patients with high tumor FAAH expression (scores 2 and 3). Similar observations were made after the analysis of *FAAH* mRNA levels in several public DNA microarray datasets[15,22,23] (Fig. 2c and Supplementary Figs. 3a–c). Interestingly, low tumor *FAAH* expression was linked to BC lung metastasis but not bone or brain metastasis[24–26] (Supplementary Figs. 3d–f). Prognostically relevant associations were not found for the other main endocannabinoid-degrading enzyme, monoacylglycerol lipase (MAGL), whose mRNA expression was analyzed by using the bc-GenExMiner website[27] (Supplementary Fig. 4).

Prognostic associations inferred from tumor data including all intrinsic BC subtypes may be biased considering that FAAH has a predominant expression in the luminal subtype of BC, which is the one with the highest inherent survival rates. To rule out the possibility that our conclusions were affected by this limitation, we next analyzed FAAH protein expression in a second TMA (i.e., TMA #2) containing primary non-treated breast tumor samples exclusively of the luminal subtype (*n* = 276) and matched synchronous lymph node (LN) metastatic samples. FAAH expression was scored as in Fig. 1c and, as expected due to the luminal nature of the samples, it was clearly detectable in most specimens, with moderate to high signal in 65% of them (Supplementary Fig. 5a). Here, low FAAH expression in

both the primary tumor and the LN samples was associated with lower overall survival, indicating that the prognostic value of FAAH is independent of their association to luminal BC (Supplementary Fig. 5b, c). In addition, an overall decrease in cancer cell FAAH expression was found in a significant proportion of the LN metastases when compared to the corresponding primary tumors (Fig. 2d, e), suggesting FAAH downregulation is a frequent event during metastatic progression of luminal BC. Specifically, 25% of the pair-matched biopsies showed a decrease in FAAH in the metastatic sample vs. only 10% of them showing an increase, and the rest of them showed no changes (Fig. 2d, f). Moreover, patients with high FAAH expression in both the primary tumor and the LN metastasis ("FAAH high") exhibited higher survival rates than those whose expression was either invariably low or decreased in the LN metastasis vs. the primary tumor ("FAAH low/decreased") (Fig. 2g).

Together, these findings show a strong association between low tumor FAAH expression (either intrinsic or acquired during metastatic progression) and high tumor aggressiveness in BC. In further support of this idea, FAAH immunostaining was absent from BC cells at the invasion front as well as in cells that had detached from the primary tumor (Supplementary Fig. 5d), which constitute the most invasive (and therefore aggressive) subpopulation of BC cells.

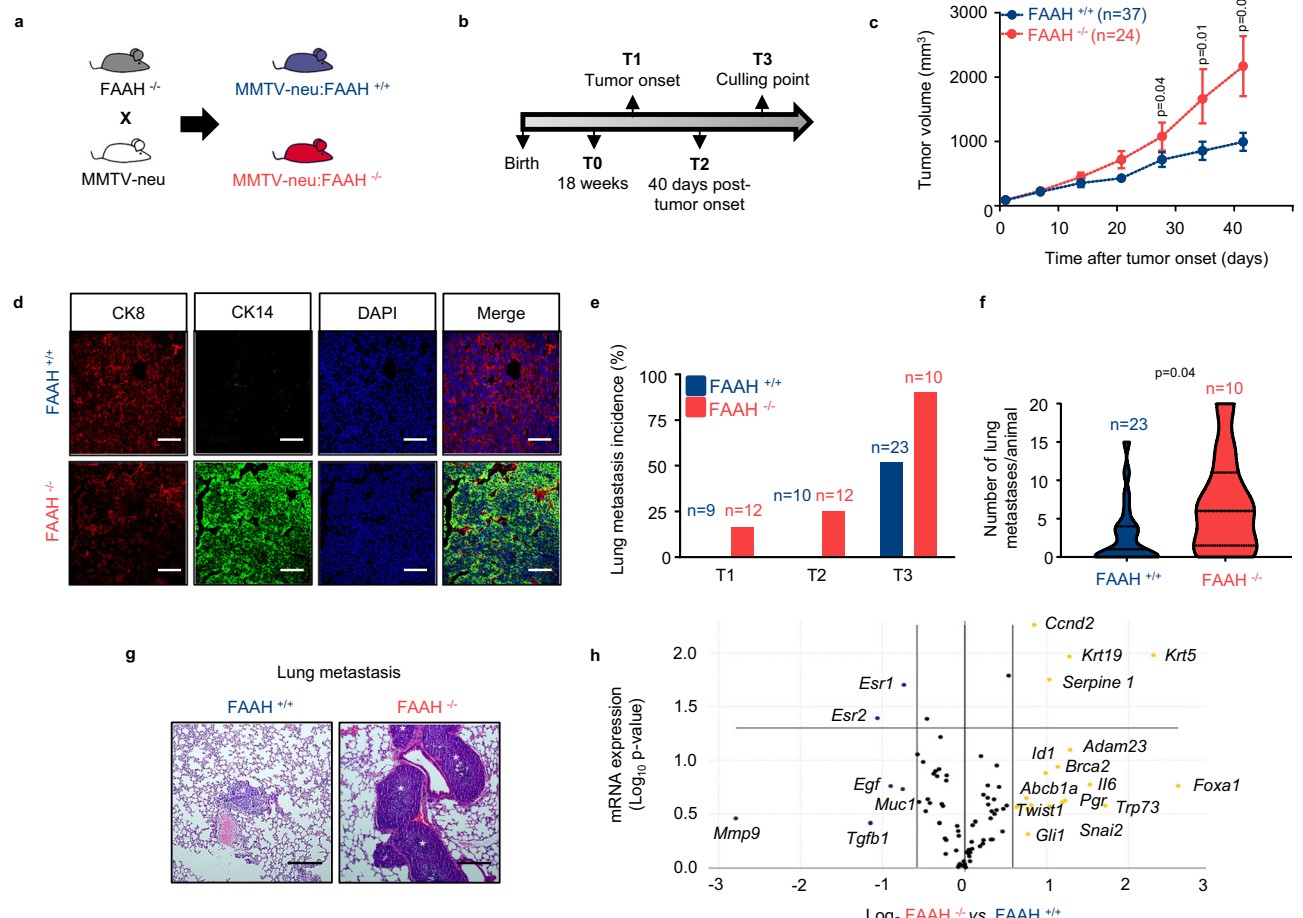

**Fig. 3 | FAAH genetic silencing promotes tumor progression and lung metastasis in a mouse model of spontaneous breast tumor formation.**
**a, b** Experimental setup. **c** Tumor growth 50 days after first tumor arousal. Data were obtained from T2 and T3 mice. Data are presented as mean values ± SEM and were analyzed by two-tailed Student's t-test. **d** Immunofluorescence analysis of MMTV-neu-derived tumors for cytokeratin 8 (CK8, red) and cytokeratin 14 (CK14, green). Data obtained from T3 mice (n = 3 biological replicates). Cell nuclei are stained in blue. Scale bar = 50 μm. **e** Percentage of animals with lung metastases at the time points established in panel b. **f** Number of metastasis per animal at T3. Data

were analyzed by 2-tailed Student's t-test. **g** Representative H&E images of lung metastases at T3 (n > 10 biological replicates). Scale bar = 250 μm. **h** Volcano plot showing data of the RT² Profiler PCR array of mouse BC. Upregulated and down-regulated genes in MMTV-neu:FAAH⁻/⁻ derived tumors with a log₂ fold change ≥0.58 (i.e., fold change ≥1.5) are depicted in yellow and blue, respectively, and statistically significant genes are over the horizontal line at −log₁₀ p value = 1.3 (i.e., p = 0.05). Each point is the result of the mean of n = 3 biological replicates mice at T3. Source data are provided as a Source Data file.

## Genetic inactivation of FAAH promotes breast tumor growth and lung metastasis in mice

To analyze whether there is a cause-and-effect link between FAAH expression and reduced BC aggressiveness, we modulated the expression of FAAH in a well-established animal model of metastatic BC: the MMTV-neu mouse model. This strain expresses the *Neu* oncogene (the rat ortholog of *HER2*), driven by the mouse mammary tumor virus (MMTV) promoter, and develops luminal-like breast tumors and lung metastases[28,29]. MMTV-neu:FAAH⁻/⁻ mice and their corresponding MMTV-neu:FAAH⁺/⁺ controls were generated by breeding MMTV-neu mice with FAAH⁻/⁻ mice[30] (Fig. 3a). Tumor generation and progression was then compared in the two genotypes at different time points: T0 (pre-lesion mammary gland), T1 (tumor onset), T2 (40 days after tumor onset) and T3 (maximum allowable size or 90 days after tumor onset, culling point) (Fig. 3b). Although tumors formed by MMTV-neu:FAAH⁻/⁻ showed a delayed tumor onset (Supplementary Fig. 6a), they significantly increased their tumor growth rate compared to their FAAH⁺/⁺ counterparts (Fig. 3c), which was accompanied by increased levels of PCNA expression (Supplementary Fig. 6b). As a result of *FAAH* genetic deletion, the levels of AEA were higher in tumors derived from MMTV-neu:FAAH⁻/⁻ mice

than in those from FAAH⁺/⁺ controls, and, remarkably, both genotypes displayed higher AEA levels in the tumors than in the non-transformed mammary gland (Supplementary Fig. 6c). No differences were found on 2-AG levels (Supplementary Fig. 6c). Histological analysis of early detected tumors (T1) revealed that MMTV-neu:FAAH⁺/⁺ mice developed low-grade adenocarcinomas, while FAAH⁻/⁻ mice developed solid high-grade carcinomas (Supplementary Fig. 6d). In more advanced stages of tumor progression (T2 and T3), both groups developed solid carcinomas with necrotic areas (Supplementary Fig. 6d). At the molecular level, MMTV-neu:FAAH⁺/⁺ tumors showed a homogeneous luminal pattern (positive for cytokeratin 8), as previously described in the literature[31], whereas MMTV-neu:FAAH⁻/⁻ tumors consisted of different components with some luminal, but mostly basal (positive for cytokeratin 14) identity (Fig. 3d). Moreover, FAAH deficiency produced a remarkable increase in the metastatic rate. Thus, while metastases in MMTV-neu:FAAH⁺/⁺ females were only evident at T3, MMTV- neu:FAAH⁻/⁻ animals showed metastatic invasion as early as in T1, when the primary tumors had just arisen (Fig. 3e). Moreover, the number of metastases per animal was significantly higher in FAAH-deficient animals at T3 (Fig. 3f, g).

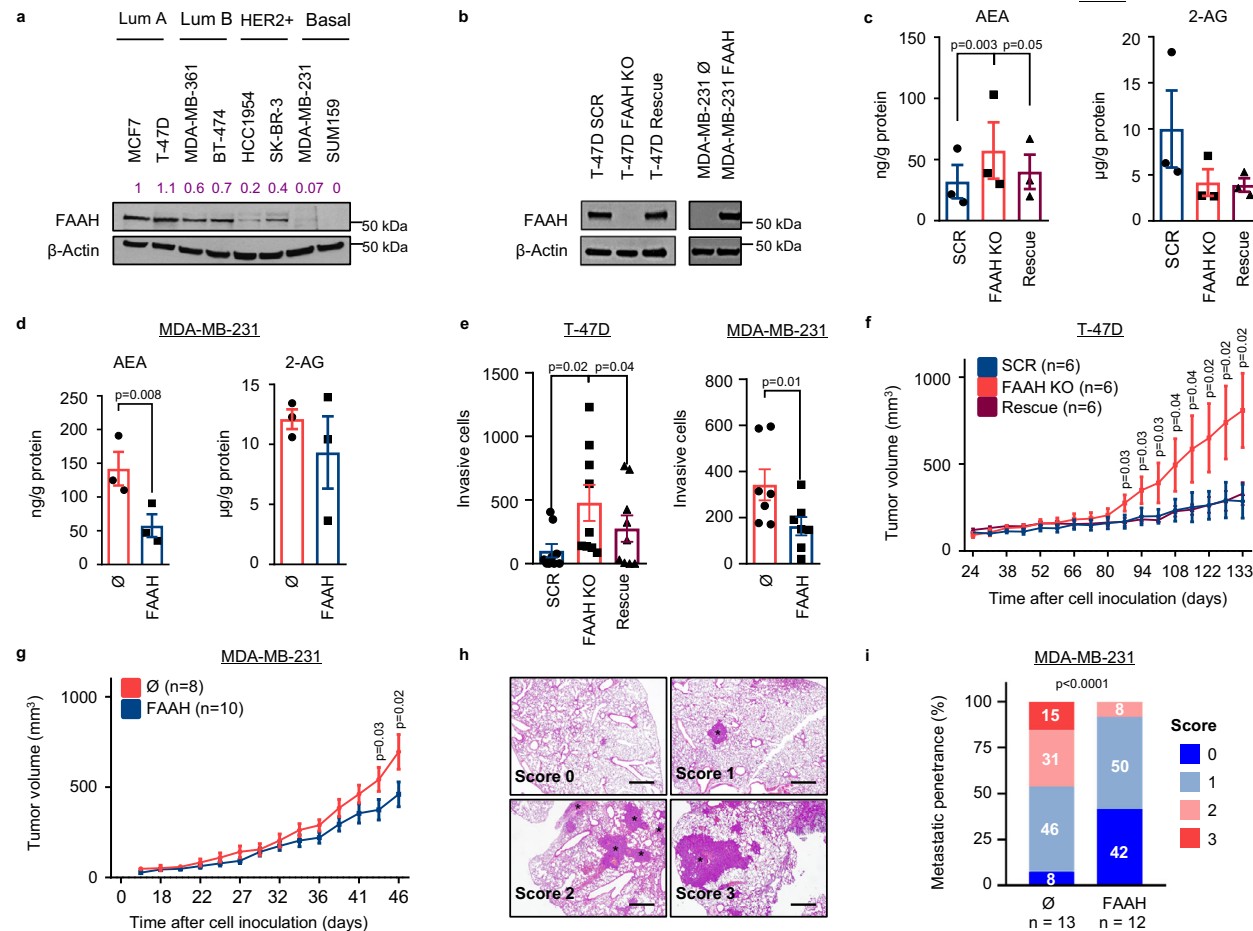

**Fig. 4 | FAAH regulates pro-oncogenic features of BC cells in vitro and in vivo.**
**a** Representative WB analysis of FAAH in a panel of human BC cell lines representative of the main intrinsic subtypes. Densitometric values after normalization against β-Actin and control condition are depicted in purple. $n = 3$ biological replicates. **b** Representative WB analysis of FAAH in FAAH-modulated T-47D and MDA-MB-231 cell lines. $n = 3$ biological replicates. **c, d** Intracellular levels of AEA and 2-AG normalized to protein content. Data are presented as mean values ± SEM of $n = 3$ biologically independent samples and were analyzed by 1-way ANOVA with post-Tukey's multiple comparison test (T-47D cells) and by two-tailed Student's t-test (MDA-MB-231). **e** BC cell invasion through matrigel in Boyden chamber assay. 10% FBS was used as chemoattractant. Data are presented as mean values ± SEM of $n = 9$ and 7 biological replicates for T-47D and MDA-MB-231 cells, respectively, and were analyzed by 1-way ANOVA with post-Tukey's multiple comparison test (T-47D

cells) and by 2-tailed Student's t-test (MDA-MB-231). **f, g** Tumor growth in athymic female mice xenografted with FAAH-modulated T-47D (**f**) and MDA-MB-231 (**g**) cell lines, respectively. Data are presented as mean values ± SEM and were analyzed by 1-way ANOVA with post-Tukey's multiple comparison test (T-47D cells) and by two-tailed Student's t-test (MDA-MB-231). **h** Representative H&E images of lung histology ($n > 10$ biological replicates) showing scoring of metastatic penetrance (% of lung parenchyma invaded by cancer cells) after injection of FAAH-modulated MDA-MB-231 cells in the tail vein of athymic female mice. 0 = no metastasis; 1 = <15%; 2 = 15–30%; 3 = 30–60%. Metastatic nodules are indicated by asterisks. Scale bar = 250 μm. **i** Percentage of metastatic penetrance in the experiment described in h. Pearson's chi-squared test was used for statistical analysis. Source data are provided as a Source Data file.

To shed some light on the molecular mechanisms of FAAH action on BC progression, we analyzed the expression of a panel of 84 BC-associated genes in MMTV-neu:FAAH$^{+/+}$ and $^{-/-}$ mice using the commercial RT² Profiler™ PCR Array of Mouse Breast Cancer (see Methods). Supporting our previous data showing an association between FAAH and luminal BC phenotypes, the expression of both isoforms of ER (*Esr1* and *Esr2*) was downregulated in tumors from MMTV-neu:FAAH$^{-/-}$ mice compared to FAAH$^{+/+}$ controls, while the basal marker cytokeratin 5 (*Krt5*) was upregulated (Fig. 3h). In addition, several genes known to be markers of poor prognosis in BC (*Twist1, Snai2, Adam23, Abcb1a, Id1*) showed an expression-upregulation trend in MMTV-neu:FAAH$^{-/-}$ tumors, consistent with their more aggressive behavior (Supplementary Data 1).

Together, these findings strongly support that FAAH plays an important role in suppressing breast tumor progression and lung metastasis in mice.

## FAAH inhibits pro-oncogenic features of human breast cancer cells

To better understand how FAAH controls tumor progression, we performed a battery of experiments in human cell models of BC. The analysis of *FAAH* mRNA expression in a panel of BC cell lines[32] revealed that, in line with our previous observations, *FAAH* levels were higher in luminal than in basal cell lines (Supplementary Fig. 7a). Similar results were obtained regarding FAAH protein expression (Fig. 4a). Based on these observations, we aimed to downregulate the expression of FAAH in a luminal A (highly FAAH-expressing) cell line and to overexpress it in a basal (low FAAH-expressing) cell line to analyze the functional impact of these modifications on cellular capabilities related to tumor progression. Specifically, we knocked out FAAH in T-47D luminal cells by using CRISPR-Cas9 technology (T-47D FAAH KO cell line) and then re-expressed it with a lentiviral vector (T-47D rescue cell line); on the other hand, we transiently overexpressed FAAH in MDA-MB-231 basal cells with a lentiviral vector (MDA-MB-231 FAAH cell line) (Fig. 4b). As

expected, the levels of AEA, but not of 2-AG, were lower in those cells endogenously or ectopically expressing FAAH (i.e., parental/rescue T-47D cells and FAAH-overexpressing MDA-MB-231 cells, respectively) than in those lacking the enzyme (i.e., T-47D FAAH KO and parental MDA-MB-231 cells) (Fig. 4c, d). Visually, parental/rescue T-47D cells had a cuboidal aspect and were densely packed in an ordered cobblestoned pattern, while T-47D FAAH KO cells had a more fusiform morphology and appeared in disordered patterns (Supplementary Fig. 7b). Consistent with that, immunofluorescence analysis revealed that FAAH silencing in T-47D cells led to a diminished expression of epithelial markers and an increased expression of mesenchymal/progression markers (Supplementary Fig. 7c), while FAAH overexpression in MDA-MB-231 cells led to the opposite phenotype (Supplementary Fig. 7d).

The absence of FAAH in BC cells, either constitutive or induced by genetic silencing, was associated with increased capabilities intimately related to tumor progression such as cell invasion (Fig. 4e) and mammosphere formation (Supplementary Fig. 7e). Interestingly, and in line with our previous observations associating lower FAAH expression to undifferentiated tumor phenotypes, we detected lower *FAAH* mRNA levels in mammospheres derived from parental T-47D cells than in their adherent, more differentiated counterparts (Supplementary Fig. 7f). Supporting these findings, *FAAH* mRNA expression was downregulated in BC subpopulations enriched in putative tumor-initiating cells (CD44$^+$/CD24$^-$)[33] (Supplementary Fig. 7g).

Next, we aimed at validating our in vitro observations in an in vivo setting. Thus, the growth of T-47D-derived xenografts in immuno-compromised mice was significantly enhanced in the absence of FAAH, and this effect was prevented by re-expressing the enzyme in the rescue cell line (Fig. 4f). In the same direction, tumor growth was significantly reduced when xenografts were generated from FAAH-overexpressing MDA-MB-231 cells compared to those generated by the parental cell line (Fig. 4g). Notably, the tumors generated by FAAH-overexpressing MDA-MB-231 cells that reached the higher volumes were found to have silenced FAAH expression to some extent, as demonstrated by WB analysis (Supplementary Fig. 7h), thus suggesting that the outcome (in terms of tumor growth) is dependent on FAAH expression.

Finally, MDA-MB-231 cells educated to metastasize to the lungs in which FAAH had been overexpressed were much less efficient at generating lung metastases upon intravenous tail injection than parental cells (Fig. 4h, i). Specifically, more than 40% of the animals injected with FAAH-overexpressing MDA-MB-231 cells were metastasis-free (compared to only 8% of the animals injected with parental MDA-MB-231 cells), and none of them reached score 3 of metastatic penetrance (compared to 15% of those in the parental MDA-MB-231 group) (Fig. 4i).

Collectively, these data demonstrate that FAAH blocks pivotal pro-oncogenic features in human BC cell lines.

## FAAH reduces pro-metastatic gene signatures in breast cancer cells

To try to unravel the molecular mechanisms responsible for FAAH-driven suppression of BC cell pro-oncogenic traits, we performed RNA-sequencing (RNA-seq) analysis in the three T-47D-derived cell lines (SCR, FAAH KO, and rescue) (Fig. 5a). In this analysis, we were interested in those genes differentially expressed in the FAAH KO cells vs. parental cells that also recovered their original levels when FAAH was re-expressed (rescued cells). We identified 491 differentially expressed genes (DEGs) in FAAH KO cells compared to the parental and rescue groups, with a fold change >1.5 and statistical significance set at $p < 0.05$. (Fig. 5b and Supplementary Data 2; some of the DEGs known to play important roles in BC progression have been highlighted in the figure). Gene Ontology (GO) pathway enrichment analysis was then carried out to functionally categorize the DEGs into three major categories: Cellular Component (CC) (Fig. 5c), Molecular Function

(MF) (Supplementary Fig. 8a), and Biological Process (BP) (Supplementary Fig. 8b). Consistent with their more mesenchymal and invasive phenotype, GO CC analysis showed that DEGs in T-47D FAAH KO cells belonged to categories involved in cell morphology, cell adhesion, directional cell migration, and extracellular matrix (ECM) reorganization (Fig. 5c). The Kyoto Encyclopedia of Genes and Genomes (KEGG) pathway analysis further identified "ECM-receptor interaction" and "focal adhesion" among the signaling pathways most impacted by FAAH knocking-out (Supplementary Fig. 8c). Supporting GO and KEGG analyses, gene set enrichment analyses (GSEA)[34] also revealed that the upregulated DEGs in T-47D FAAH KO cells were significantly associated to highly aggressive signatures such as those of epithelial-to-mesenchymal transition (EMT)[35], BC metastasis to the lungs[36] (Supplementary Figs. 8d, e), basal and mesenchymal traits[37], MaSC populations[38], and van't Veer poor prognosis BC signature[25] (Fig. 5d), which is used clinically to identify BC patients at a greater risk of developing metastatic disease. All these data point to FAAH as an inhibitor of critical cellular processes in cancer cell migration and invasion, and therefore, in metastasis.

To corroborate the molecular association between FAAH and BC metastasis, we analyzed the mRNA expression of a panel of 84 metastasis-associated genes in MDA-MB-231 parental cells and FAAH-overexpressing MDA-MB-231 cells using the commercial RT$^2$ Profiler™ PCR Array of Human Tumor Metastasis (see Methods) (Fig. 5e and Supplementary Data 3). Most of the genes included in the PCR array (83%, 70 out of 84) appeared downregulated in FAAH-overexpressing MDA-MB-231 cells compared to parental MDA-MB-231 cells (Supplementary Data 3). Among them, those with a well-established role in BC metastasis are depicted in Fig. 5e.

Together, these data further support that FAAH reduces the aggressiveness of BC cells by inhibiting the expression of invasion and metastasis-related genes.

## The CXCR4-CXCL12 axis is involved in FAAH-mediated inhibition of breast cancer progression

Among all the candidate genes that were significantly deregulated upon FAAH modulation in both T-47D and MDA-MB-231 cells, chemokine receptor CXCR4 was selected for functional validation owing to its well-established role in lung-specific metastasis of BC[39]. The RNA-seq and PCR array data revealed that *CXCR4* mRNA levels were higher in cells with low FAAH expression (T-47D FAAH KO and parental MDA-MB-231 cells) than in their FAAH-expressing counterparts (T-47D SCR/rescue and FAAH-overexpressing MDA-MB-231 cells, respectively) (Fig. 5). These observations were confirmed at the protein level after flow cytometric analysis of CXCR4 in the cell membrane of T-47D and MDA-MB-231 sublines (Fig. 6a). The inverse association between CXCR4/CXCL12 and FAAH expression was also observed in mouse and human tumors. Thus, breast tumors generated by MMTV-neu:FAAH$^{-/-}$ mice expressed higher mRNA (Supplementary Fig. 9a) and protein (Fig. 6b) levels of *Cxcr4* and *Cxcl12* than their FAAH-expressing counterparts, which is consistent with their enhanced aggressive behavior and metastatic rate. In addition, the study of 1904 human BC samples from public DNA microarrays[15] showed an inverse correlation between *FAAH* and *CXCR4*/*CXCL12* mRNA levels in human tumors (Fig. 6c), and GSEA analysis of the 286 human BC samples included in the Wang dataset[26] revealed an activation of the CXCR4 pathway[40] in human BC tumors with low FAAH expression (Supplementary Fig. 9b).

We then aimed to determine whether there was a causal link between FAAH-dependent modulation of the invasive phenotype and the regulation of the CXCR4-CXCL12 signaling axis in T-47D and MDA-MB-231 cells. Blockade of CXCR4 with the selective antagonist AMD3100 prevented cell invasion only in cells with no FAAH expression (i.e., T-47D FAAH KO and parental MDA-MB-231 cells), indicating that upregulation of CXCR4 in these cells is (at least partially) responsible for their enhanced invasive capacity (Fig. 6d, e). In

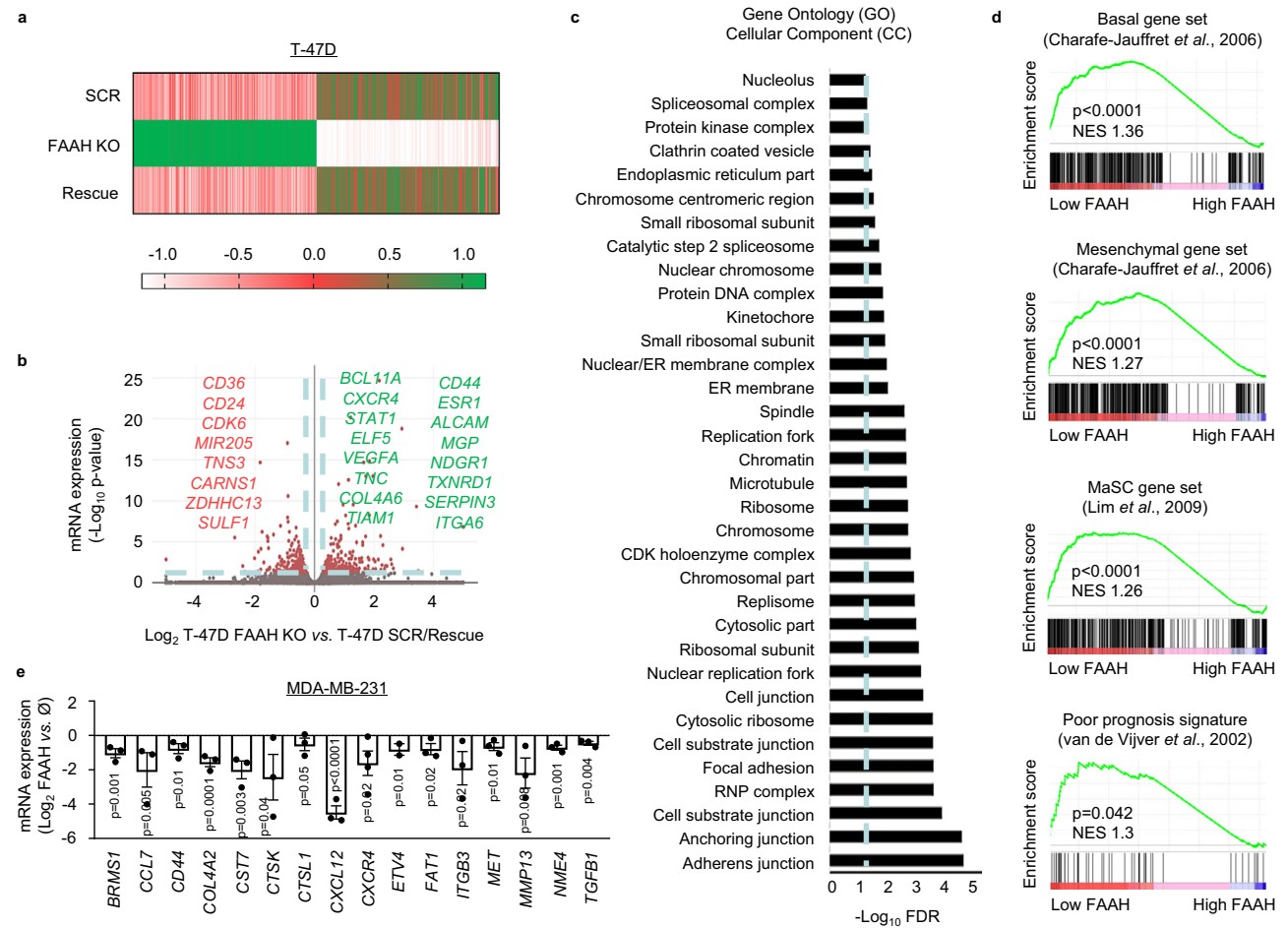

**Fig. 5 | FAAH modulation in BC cells alters the levels of invasion and metastasis-related genes. a** Heatmap showing fold-change expression of the differentially expressed genes (DEGs = 491) in the RNA-seq of FAAH-modulated T-47D cells. DEGs considered in this analysis were only those significantly deregulated in T-47D FAAH KO cells vs. parental cells that also recovered their original levels in the rescue cells. **b** Volcano plot showing some of the top DEGs (downregulated in red, upregulated in green) in T-47D FAAH KO cells compared to parental/rescued cells. Dashed blue lines establish inclusion cut-off values at $-\log_{10}$ false discovery rate (FDR) $\geq$1.3 (i.e., FDR $\leq$0.05) and $\log_2$ fold change $\geq$0.58 (i.e., fold change $\geq$1.5). Some relevant upregulated and downregulated DEGs are shown in green and red, respectively.

**c** Bar graph representing Gene Ontology (GO) Cellular Component (CC) analysis of DEGs in T-47D FAAH KO cells compared to parental/rescue cells. **d** Gene Set Enrichment Analysis (GSEA) of DEGs in T-47D FAAH KO cells compared to parental/rescue cells. **e** Bar graph showing results of the RT$^2$ Profiler PCR array of human tumor metastasis in FAAH-modulated MDA-MB-231 cells. The expression of BC metastasis-related downregulated genes in FAAH-overexpressing MDA-MB-231 cells is depicted as $\log_2$ fold change vs. parental cells. Data are presented as mean values $\pm$ SEM of $n = 3$ biological replicates and were analyzed by two-tailed Student's $t$-test. Source data are provided as a Source Data file.

addition, FAAH overexpression in MDA-MB-231 cells decreased their ability to invade toward CXCL12 (Fig. 6e).

These findings support a pivotal role of the CXCR4-CXCL12 axis in FAAH-mediated inhibition of the protumorigenic traits of BC cells.

## FAAH suppresses pro-invasive capabilities of breast cancer cells by decreasing the anandamide tone

AEA is the biologically most relevant FAAH substrate and, accordingly, BC cell lines displayed significant changes in their intracellular AEA levels upon FAAH modulation (Fig. 4c). Hence, we aimed to determine whether the AEA tone, as driven by FAAH, is the responsible cue for the control of the protumoral features of BC cells described above. Western blot analysis revealed that metAEA, a non-hydrolyzable structural analog of AEA, upregulated CXCR4 expression in T-47D cells (Fig. 7a), thus mimicking the effect of FAAH genetic ablation. This effect was mediated, at least in part, by the activation of CBRs as it was partially prevented by co-incubation of the cells with the CB$_1$R- and CB$_2$R-selective antagonists SR141716 (SR1) and SR144528 (SR2), respectively (Fig. 7a). In the same line, metAEA induced an EMT-like phenotype (downregulation of E-cadherin and upregulation of vimentin) in T-47D cells, similar to

that found upon knocking-out FAAH, and this was also partially prevented by SR1 and SR2 (Supplementary Fig. 10a).

MetAEA also phenocopied the effect of FAAH knocking-out on the CXCR4-dependent invasive capacity of T-47D cells. Thus, metAEA increased invasion of T-47D parental cells through CBR activation (as demonstrated by the prevention by CB$_1$R or CB$_2$R blockade), and this effect was prevented by blocking CXCR4 with AMD3100 (Fig. 7b). In the same direction, metAEA rescued the decreased invasion displayed by FAAH-overexpressing MDA-MB-231 cells also in a CBR- and CXCR4-dependent manner, suggesting that a downregulation of endogenous AEA upon FAAH overexpression in MDA-MB-231 cells is involved in their less invasive behavior (Fig. 7c). Finally, metAEA was demonstrated to have a prometastatic role in vivo: immunodeficient mice injected with lung-tropic, FAAH-overexpressing MDA-MB-231 cells that were subsequently treated with metAEA showed similar metastatic burden than mice injected with parental (i.e., FAAH low-expressing) cells (Fig. 7d). This metastasis-boosting effect was completely prevented by co-treatment with SR1, and partially prevented by co-treatment with SR2 or AMD3100, thus indicating that metAEA exerts its prometastatic action in vivo, at least in part, through the activation

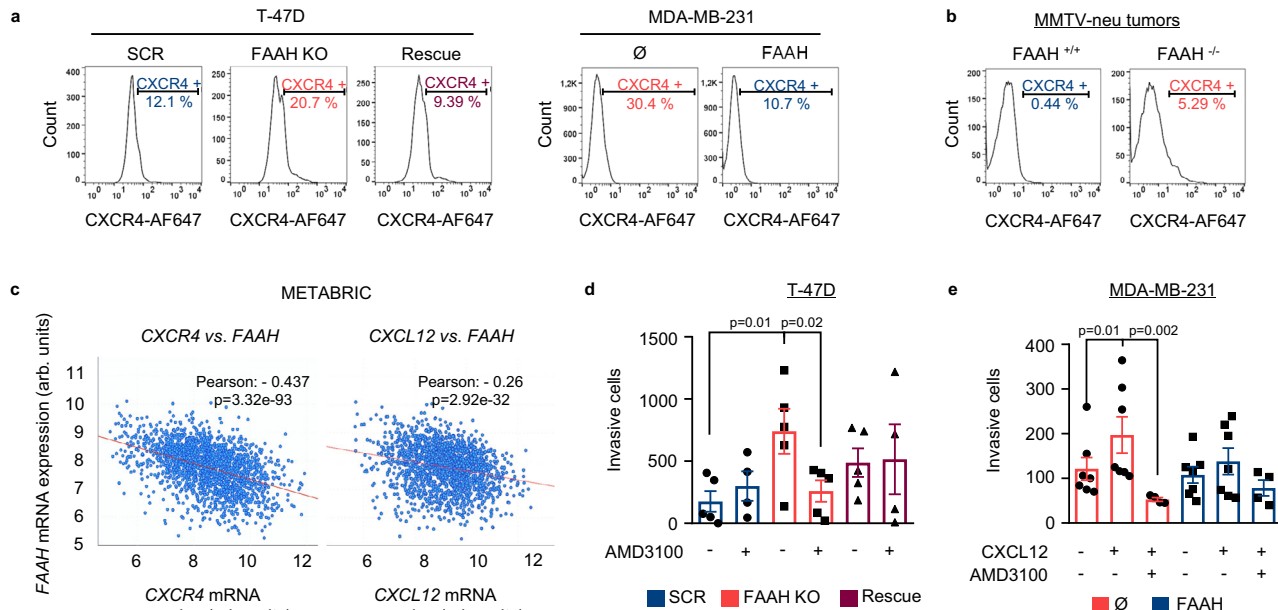

**Fig. 6 | The CXCR4-CXCL12 axis is involved in the control of FAAH-mediated inhibition of the invasive phenotype of BC cells.** Representative histograms of flow cytometric detection of cell membrane CXCR4 in FAAH-modulated T-47D and MDA-MB-231 cell lines (**a**) and in tumors derived from MMTV-neu:FAAH[+/+] and FAAH[−/−] mice (**b**). **c** Scatter plot showing the inverse correlation between *FAAH* and *CXCR4* or *CXCL12* mRNA expression in BC samples according to the METABRIC dataset[15]. Correlative associations were calculated by the Pearson's r. Analysis of AMD3100 treatment (**d**, **e**) and CXCL12-induced chemotaxis (**e**) on the invasive capacity of FAAH-modulated T-47D (**d**) and MDA-MB-231 (**e**) cells. AMD3100 (1 μM) was added to the upper compartment after cell seeding. CXCL12 (50 nM) was added to the lower compartment of the Boyden chamber containing serum-free medium. Chemotactic effect of CXCL12 in T-47D cells could not be tested due to their null invasivity towards serum-free medium. Data are presented as mean values ± SEM of *n* = 5 and 7 biological replicates for T-47D and MDA-MB-231 cells, respectively, and were analyzed by 1-way ANOVA with post-Tukey's multiple comparison test. Source data are provided as a Source Data file.

of CBRs and CXCR4 (Fig. 7d). Importantly, none of the antagonists exerted an antitumoral effect when used alone (Supplementary Fig. 10b). Consistent with all of the aforementioned data, we found that higher AEA levels in human tumor samples were significantly associated with a lower overall survival of the patients (Fig. 7e). Together, these findings support the existence of an endogenous AEA tone that, acting through CBRs and ultimately controlled by FAAH activity, induces protumorigenic and prometastatic phenotypes in BC cells.

## Discussion

During the last two decades, the use of cannabinoid agonists has proven to exert antitumor actions in different cancer models, including BC[10]. Although this implies the existence of a functional endocannabinoid system in tumor cells, very few studies have addressed the role of this system in the generation, proliferation, and progression of cancer. An aberrant expression of the endocannabinoid-degrading enzyme FAAH has been reported in different cancer models compared to their non-transformed counterpart[6], however, its contribution to tumor progression is not clear yet. Here, we aimed at investigating the functional role of FAAH in BC.

First, we demonstrate that FAAH is highly expressed in luminal BC, and that its downregulation in the primary tumor and/or LN metastasis is a marker of poor prognosis. Approximately 80% of BC patients are now diagnosed as ER+, and almost all of these are prescribed 5 years of adjuvant endocrine therapy[41]. Although such treatment markedly reduces mortality, long-term recurrence (>5 years) remains a major problem[3], as tumors frequently recur up to two decades after initial diagnosis with visceral metastasis[4,5]. We propose that assessing FAAH levels in the tumor may be useful to identify those luminal BC patients that may benefit from more aggressive treatments to decrease the risk of metastatic relapse. However, the actual translational potential of FAAH as a valuable prognostic marker would require, aside from validation in larger cohorts of patients, the demonstration that it performs better than the currently used prognostic platforms such as Mammaprint or Oncotype DX.

Second, we show that FAAH plays an important role in the control of the luminal phenotype of BC cells. In human tumors and BC cell lines, FAAH expression was strongly associated with luminal features such as HR+ status and high differentiation grade, which is in line with previous observations by Jessani and cols., who detected FAAH activity exclusively in luminal BC lines MCF7 and T-47D, but not in basal-like BC cell lines such as MDA-MB-231[42]. Additionally, FAAH genetic silencing in mouse and cell models of luminal BC promoted a phenotypic shift towards more aggressive and undifferentiated tumors with basal-like features, while FAAH overexpression in basal-like BC cells attenuated aggressive behavior. Together, these results show that FAAH is not just a marker, but also a driver of a less aggressive, luminal-like tumor phenotype, and therefore it offers the potential to be studied as a candidate for differentiation therapy in patients with basal-like BC.

Third, our findings point to a pivotal role of FAAH in suppressing BC metastasis. Thus, FAAH genetic inactivation in MMTV-neu mice increased the rate of spontaneous lung metastasis, while expression of FAAH in lung-seeking MDA-MB-231 cells[36] was sufficient to inhibit the formation of lung metastases after tail-vein injection in immunocompromised mice. Organotropic metastasis of BC is attributed to the early acquisition of specific gene expression signatures that bestow cancer cells with metastatic capabilities toward selective sites[43]. Unlike other metastatic signatures, which include genes functionally critical for the host organ (for example, the bone metastasis signature[44] encodes proteins that alter the bone tissue environment to foster the formation of osteolytic bone lesions), the lung metastasis signature (LMS)[36] is less specific to the lung microenvironment and rather promotes general features of aggressive growth and invasiveness in the cancer cells such as ER- status, a Rosetta-type poor prognosis signature[23], and a basal-like phenotype. Our data suggest that FAAH impairs both BC progression and lung metastasis by blocking all those aggressive features.

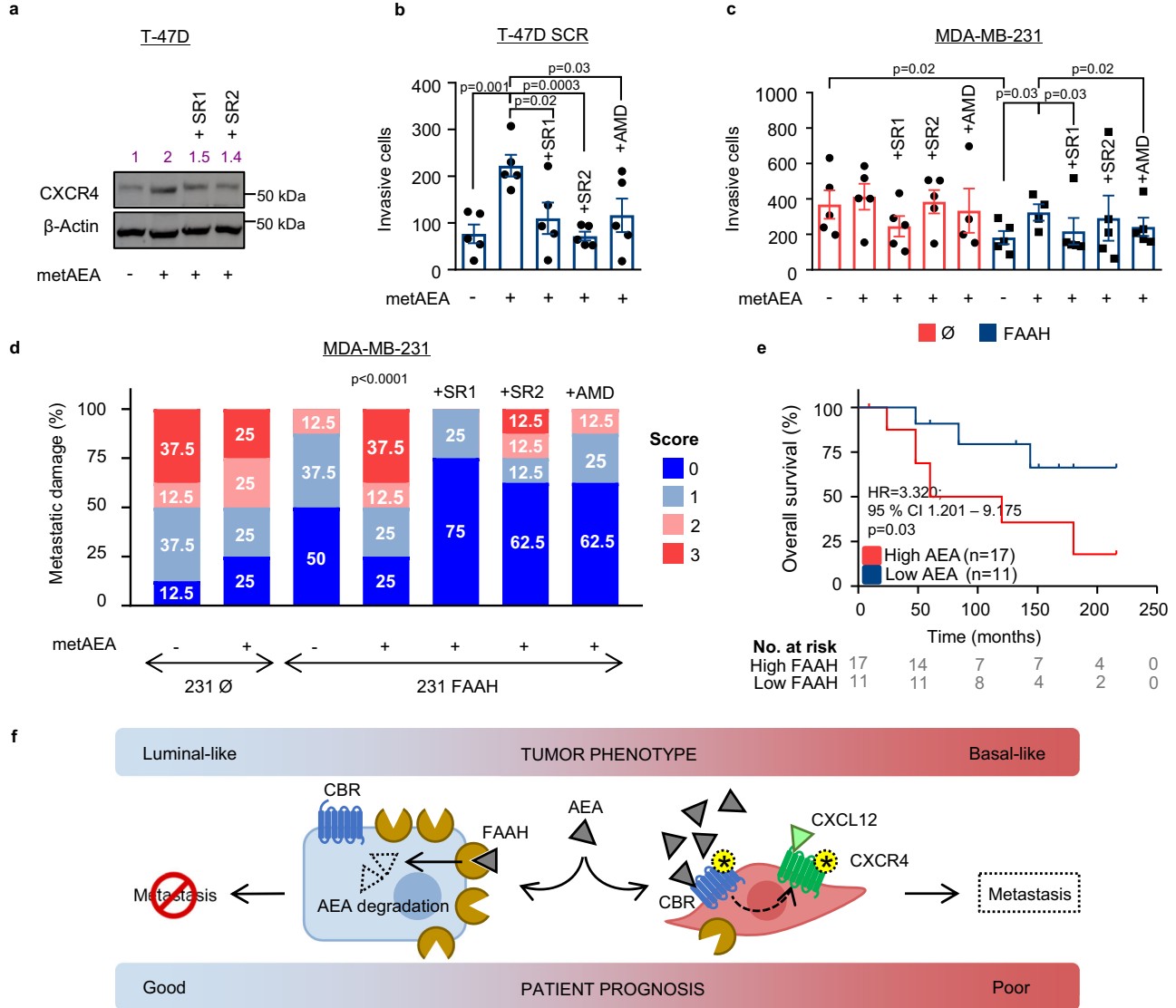

**Fig. 7 | metAEA enhances CXCR4-mediated pro-invasive capacity of human BC cell lines through a CBR-mediated mechanism. a** Representative WB analysis of CXCR4 in T-47D parental cells after being treated with metAEA (1 nM) alone or in combination with SR1 or SR2 (1 μM) for 24 h. Antagonists were added to the cells 1 h before metAEA. Densitometric analysis is shown in purple. *n* = 3 biological replicates. Invasion of T-47D (**b**) and MDA-MB-231 (**c**) under the treatment of metAEA (1 nM) alone or in combination with SR1, SR2, or AMD3100 (1 μM) as determined in matrigel-coated Boyden chambers. Antagonists were added to the upper compartment 1 h before metAEA. Data are presented as mean values ± SEM of *n* = 5 biological replicates and were analyzed by 1-way ANOVA with post-Tukey's multiple comparison test. **d** Lung metastatic damage caused by parental and FAAH-overexpressing MDA-MB-231 cells in immunocompromised mice treated with metAEA (0.1 mg/kg) alone or in combination with SR1, SR2 or AMD3100 (2.5 mg/kg). Metastatic damage was monitored with an IVIS system and categorized into four scores depending on bioluminescence values. Pearson's chi-squared test was used for statistical analysis. **e** Kaplan-Meier curves for overall survival of BC patients with high and low AEA levels. Kaplan-Meier curves were statistically compared by the log-rank test. **f** Schematic drawing of the proposed metastasis-suppressor mechanism of FAAH in BC cells. On the left, high FAAH expression determines low anandamide (AEA) tone and a luminal-like phenotype. On the right, low FAAH expression allows higher AEA levels that, acting through cannabinoid receptors (CBR), induce the expression of CXCR4 and CXCR4-mediated pro-metastatic responses. Yellow asterisks represent activated receptors. Source data are provided as a Source Data file.

More specifically, we provide evidence for the functional involvement of one of the members of the LMS signature, the chemokine receptor CXCR4, in FAAH-mediated anti-metastatic phenotype. Activation of CXCR4 in BC cells by its ligand, CXCL12, promotes chemotaxis and migration of cancer cells towards CXCL12-rich environments, which coincidentally represent the main sites for BC metastasis, namely the bones, lymph nodes, lungs, and liver[39,45,46]. Consistent with that, CXCR4 blockade significantly impairs BC metastasis to the lungs and the lymph nodes[47–49]. Our data show that CXCR4 and CXCL12 expression are inversely associated with FAAH expression in mouse and human tumors, and that, in BC cell lines, FAAH negatively regulates CXCR4 expression and CXCR4-dependent invasive capacity.

Although regulation of BC metastasis by FAAH probably relies on multiple signaling routes, these findings strongly support an involvement of CXCR4-CXCL12 signaling in FAAH-mediated anti-metastatic behavior, and justify further research on the importance of the FAAH-CXCR4 axis in metastasis formation in vivo.

The identification of the downstream effectors of FAAH action remains an important goal to understand the role of this protein in BC. Here, we show that the treatment with the AEA analog metAEA phenocopies some of the in vitro (induction of CXCR4 expression and CXCR4-mediated cell invasion) and in vivo (induction of CXCR4-dependent lung metastasis) effects of FAAH silencing through a CBR-mediated mechanism, which strongly supports the existence of an endogenous

AEA tone in BC cells that supports tumor progression and that is ultimately controlled by FAAH. Supporting our hypothesis, several groups have previously detected higher levels of AEA in tumor samples vs. the corresponding non-transformed tissue (gliomas, meningiomas, pituitary adenomas, prostate, colon, and endometrial cancer)[6], and in plasma of patients with metastasis than in those with no metastasis[50].

We are aware that the existence of a protumorigenic AEA tone in BC may be at odds with previous studies describing the antitumoral actions of AEA (or AEA analogs) in BC cells[51–54]. However, the antitumor effects described in those studies were exerted by micromolar concentrations of the compounds, which are very far from those achieved in vivo. We (present study) and others[55] have demonstrated that AEA is present in the order of picomol per gram of protein in BC cells, which, by estimating a cellular protein content of ~20% (w/v), points to nanomolar-range levels. Therefore, we firmly believe that concentrations in the low nanomolar range reflect more accurately the endogenous AEA tone in BC cells and are therefore most appropriate to draw conclusions on the role of the endocannabinoid system in BC.

In summary, our findings show that FAAH is a robust prognostic indicator in luminal BC and a critical regulator of luminal-like phenotype and BC metastasis. Hence, therapeutic strategies could be developed for the treatment of this disease with FAAH as a direct or indirect target. Specifically, our data suggest that BC patients would benefit from an upregulation of FAAH expression and/or activity in the cancer cells, but due to the current technical limitations of gene therapy, a more rational approach to bring FAAH upregulation closer to the clinic would be to identify unequivocal upstream regulators that increase its expression or activity. Recently, the first compound ever reported to stimulate FAAH was isolated from *Arabidopsis thaliana*, thus providing a tool to downregulate the AEA tone in the cell[56]. Future studies may focus on the signaling pathways controlled by FAAH in the context of advanced BC, which could pave the way for the design of strategies and tools for the management of this malignant disease.

## Methods

### Animals
All procedures involving animals were performed with the approval of the Complutense University Animal Experimentation Committee and Madrid Regional Government according to the European official regulations. Animals were housed in the animal facility of the UCM School of Biology, under a 12 h light-dark cycle, and were allowed to feed and drink *ad libitum*.

### Generation of MMTV-neu:FAAH$^{-/-}$ mice and sample collection
Generation of the congenic strain MMTV-neu:FAAH$^{-/-}$ was accomplished by mating MMTV-neu mice (The Jackson Laboratory, Bar Harbor, ME, US) with FAAH$^{-/-}$ mice (kind donation by Dr. Ben Cravatt's laboratory, Scripps Research Institute, La Jolla, CA, US). To transfer the FAAH$^{-/-}$ line (with a C57BL/6 J background) to the genetic background of the tumor-prone animals (FVB/NJ), the descendants were backcrossed with MMTV-neu mice for 6 generations, using a marker-assisted selection protocol (MASP) as described in ref. [57]. A total of 64 MMTV-neu:FAAH$^{+/+}$ and 58 MMTV-neu:FAAH$^{-/-}$ female mice were palpated twice weekly for mammary gland nodules and, as soon as tumors appeared, they were routinely measured with external caliper, and volume was calculated as $(4\pi/3) \times (width/2)^2 \times (length/2)$. After animal sacrifice, tumors were extracted and divided into four portions for (1) immunofluorescent staining [frozen in Tissue-Tek (Sakura Finetek), (2) H&E staining (fixed in 10% formalin), (3) protein extraction (snap frozen), and (4) RNA isolation (snap frozen). Non-transformed mammary glands and lungs were fixed in 10% formalin O/N and changed to 50% ethanol the day after. Macroscopic metastases in the lungs were visually assessed before tissue fixation and microscopic metastases were determined by subsequent H&E staining of fixed paraffin-embedded sections.

### Generation of xenografts in immunocompromised mice
$1 \times 10^7$ T-47D cells in a total volume of 200 μL (100 μL cell suspension +100 μL GFR Matrigel®) were subcutaneously injected into the right flank of 6-week-old athymic female mice (Charles Rivers Laboratories). Drinking water of these mice was supplemented with 0.67 μg/mL 17β-estradiol (Sigma-Aldrich) from 1 week prior to inoculation until sacrificed. In the case of MDA-MB-231 cells, $2.5 \times 10^6$ cells in a total volume of 50 μL were orthotopically inoculated into the right inguinal mammary fat pad of athymic mice. Tumors began to arise 4 (T-47D) or 2 (MDA-MB-231) weeks post inoculation and were monitored three times a week for growth by measuring length and width with a digital caliper. Tumor volume was calculated as $(4\pi/3) \times (width/2)^2 \times (length/2)$. When tumors reached a volume >1000 mm$^3$, mice were euthanized, and tumors were excised.

### Generation of lung metastasis and treatments in immunocompromised mice
Lung metastases were generated by injection of $5 \times 10^5$ lung-seeking MDA-MB-231$^{LUC+}$ into the lateral tail vein of 6-week-old athymic female mice. For the experiment performed in Fig. 4h, i, animals were euthanized 6 weeks after cell injection and lungs were excised, fixed in 10% formalin O/N, and changed to 50% ethanol the day after. Fixed lungs were then paraffin-embedded at the Department of Medicine and Animal Surgery at UCM School of Veterinary, which also determined the presence and extension of metastatic lesions. For the experiment performed in Fig. 7d, lung metastases were generated as described above and, starting from day 2 after cell injection, animals were treated with vehicle or with metAEA (0.1 mg/kg) alone or in combination with SR1 (2.5 mg/kg), SR2 (2.5 mg/kg), or AMD3100 (2.5 mg/kg) 3 times per week for 6 weeks through i.p. injections. All treatments were diluted in a solution composed of ethanol:Tween 80: saline at a 1:1:18 (v/v/v) ratio. Antagonists were injected 30 min before metAEA. Monitoring of metastatic lesions was performed by bioluminescence after an i.p. injection of D-Luciferin (Goldbio) at 150 mg/kg in an IVIS Spectrum system (Perkin Elmer). Imaging data were processed with Living Image® software (Perkin Elmer). Normalized bioluminescence values at week 6 were categorized into four metastatic scores that represented the occurrence of metastatic damage.

### Human samples
All studies were authorized by the respective Hospital Ethics Committees. Tissue microarrays (TMAs) were generated by taking 1 mm punches from a series of paraffin-embedded (female donor patients) tissue blocks that were transferred into a positionally encoded array in a recipient paraffin block. TMA #1 included 617 breast tumor samples of all BC subtypes from cases operated in the University Hospitals of Kiel, Tübingen, or Freiburg between 1997 and 2010, as described in ref. [58]. The mean duration of follow-up was 48.51 months, standard deviation 29.71, and the median duration of follow-up 44.0 months. TMA #2 included 276 pair-matched samples of luminal BC and affected lymph nodes from cases operated on in the University Hospital of Donostia and Onkologikoa and were donated by Dr. Alvarez-Lopez and Dr. Caffarel (Biodonostia Health Research Institute, San Sebastián, Spain). The mean duration of follow-up was 99.88 months, standard deviation 37.05, and the median duration of follow-up 105.6 months. Complete histopathological information, date, and cause of death, as well as the date of local and/or distant relapse were available for all the patients and are available in Supplementary Fig. 11. For AEA measurement, a series of 37 breast tumors of all BC subtypes from the tumor biobank of Hospital Universitario Puerta de Hierro (Majadahonda, Madrid, Spain) was used. All patients gave informed consent, and the study was authorized by the respective Hospital Ethics Committees without compensation for any of the participants.

## Analysis of transcriptomic data from public datasets

Clinically relevant associations based on the mRNA expression levels of *FAAH* and other metabolic enzymes of the endocannabinoid system were studied by using the "expression"[14] and the "prognostic"[27] modules of the BC Gene-Expression Miner (bc-GenExMiner) website. Kaplan-Meier curves showing overall or metastasis-free survival according to *FAAH* expression were generated by using the Kaplan-Meier Plotter (KM plotter) website[22]. Best threshold cutoffs were selected automatically by the program. GSEA analysis was performed with GSEA v4.3.2 (Broad Institute, Inc., Massachusetts Institute of Technology, and Regents of the University of California).

## Reagents

Methanandamide (metAEA) was purchased from Cayman Chemical. Selective antagonists of $CB_1R$ (SR141716, SR1) and $CB_2R$ (SR144528, SR2) were purchased from Tocris Bioscience. Selective antagonist of CXCR4 AMD3100 was purchased from Sigma-Aldrich. CXCR4 ligand CXCL12 was purchased from PeproTech.

## Cell cultures

Human BC cell lines T-47D and MDA-MB-231 were obtained from the American Type Culture Collection (ATCC). T-47D cells were cultured in RPMI medium (Sigma-Aldrich) with 10% fetal bovine serum (FBS, Gibco), 10 μg/mL insulin (Sigma-Aldrich) and 1% penicillin/streptomycin (Lonza). MDA-MB-231 cells were cultured in DMEM medium (Sigma-Aldrich) with 10% FBS and 1% penicillin/streptomycin. All cell lines were maintained at 37 °C in a humidified atmosphere of 5% $CO_2$ and routinely tested for *Mycoplasma* spp. contamination.

## CRISPR genome editing

A T-47D FAAH KO cell line was generated by knocking out FAAH expression with a CRISPR/Cas9 kit from Origene. Briefly, the Amaxa™ HT Nucleofector™ (Lonza) was used to co-transfect $1 \times 10^6$ cells with 1 μg of guide RNA targeting *FAAH* (CCG CCA CGA AGC AGG CC) or a scrambled guide RNA (GCA CTA CCA GAG CTA ACT CA) (SCR) vector together with 1 μg of donor DNA (which provided antibiotic resistance). Cells were passaged for an additional two weeks and then seeded at a low density in puromycin (0.5 μg/mL)-containing medium. After antibiotic selection, puromycin-resistant colonies were isolated, expanded, and analyzed by genomic PCR. The T-47D FAAH KO cell line was subsequently used to obtain the T-47D rescue cell line by lentiviral delivery of human *FAAH* as described below.

## Lentiviral transduction

For lentiviral transduction of T-47D FAAH KO and MDA-MB-231 cells, a second-generation packaging system was used. Commercial packaging (psPAX2) and envelope (pMD2.G) plasmids (Addgene), as well as the pReceiver-Lv225 (Genecopoeia), were kindly donated by Dr. Fernández-Piqueras (Centro de Biología Molecular Severo Ochoa (CBMSO) (CSIC-UAM), Madrid, Spain). EX-hFAAH-Lv225 transfer plasmid was home generated by subcloning human *FAAH* into the XhoI and BstbI sites of pReceiver-Lv225. The empty (Ø) vector EX-NEG-Lv225 was used as negative control. Briefly, lentiviruses were generated by transfecting $3 \times 10^6$ HEK-293T cells in antibiotic-free medium with 6 μg of total DNA in a 1:2:3 ratio of pMD2.G:psPAX2:transfer plasmid using polyethylenimine (Sigma-Aldrich). 48 h after transfection, conditioned medium containing recombinant lentiviruses was collected, filtered through 0.45 μm sterilization filters and added to T-47D or MDA-MB-231 cells together with 8 μg/mL polybrene (Sigma-Aldrich). Transduction efficiency was monitored by checking the expression of green fluorescent protein (GFP), which usually happened 48-72 h later. Antibiotic selection was not performed as the transfection efficiency was usually near 100%. However, while T-47D rescue cells maintained FAAH overexpression over passages, MDA-MB-231 FAAH cells usually lost it after a few weeks (even under antibiotic pressure), and therefore experiments with this cell line were always performed in transient.

## Endocannabinoid measurement

For sample preparation, cells were grown in 6 cm dishes and scraped off in ice-cold PBS. They were centrifuged twice at $2000 \times g$ for 10' at 4 °C and pellets were frozen in liquid nitrogen and stored at −80 °C until used. Lipid extraction and parallel quantification of endocannabinoids and endocannabinoid-like molecules were performed at the lipidomics unit of the *Universitätsmedizin* associated with the University of Mainz (Germany).

## RNA extraction and reverse transcription PCR (RT-PCR)

RNA was isolated with Nucleozol™ (Macherey-Nagel) following the extraction protocol suggested by the manufacturer. Reverse transcription (RT) was performed from 2 to 3 μg of RNA with the Transcriptor First Strand cDNA Synthesis Kit (Roche Life Science) using random hexamer primers. End-point RT-PCR was performed with the DreamTaq DNA Polymerase (Thermo Fisher Scientific) and results were visualized in a 1.5% agarose gel. Real-time quantitative RT-PCR (Q-PCR) was performed in 384-well plates using the LightCycler® Multiplex DNA Master (Roche Life Science) and hybridization probes from the Universal ProbeLibrary Set (Roche Life Science). Data were acquired with a QuantStudio™ 12K Flex system (Applied Biosystems). The relative expression ratio of target/reference gene was calculated with the $\Delta\Delta Ct$ method[59]. The following primers were used (5'−3' sequence; F: Forward, R: Reverse, h: Human, m: Mouse): hFAAH F TGG GAA AGG CCT GGG AAG TGA ACA, hFAAH R GCC GCA GAT GCC GCA GAA GGA G, hOCT4 F CGT GCA GGC CCG AAA GAG AAA G, hOCT4 R CTG CTG GGC GAT GTG GCT GAT, hSOX2 F AAC ATG ATG GAG ACG GAG CTG AAG, hSOX2 R TAC GCG CAC ATG AAC GGC TGG AG, hTBP F CCC ATG ACT CCC ATG ACC, hTBP R TTT ACA ACC AAG ATT CAC TGT GG, mCxcr4 F TGG AAC CGA TCA GTG TGA GT Cxcr4 R GGG CAG GAA GAT CCT ATT GA, mCxcl12 F CTG TGC CCT TCA GAT TGT TG, mCxcl12 R TAA TTT CGG GTC AAT GCA CA, mTbp F GGG GAG CTG TGA TGT GAA GT, mTbp R CCA GGA AAT AAT TCT GGC TCA.

For pathway-focused gene expression analysis, SYBR® Green-based Q-PCR was performed using the RT² Profiler PCR Array System (Qiagen) in accordance with the manufacturer's guidelines. For MMTV-neu tumors, the RT² Profiler™ PCR Array Mouse Breast Cancer (#PAMM-131Z) was used. For FAAH-modulated MDA-MB-231 cells, the RT² Profiler™ PCR Array Human Tumor Metastasis (#PAHS-028Z) was used.

## RNA sequencing (RNA-seq)

RNA was isolated as described above and resuspended in ultrapure water. Concentration and RNA integrity number >7 were determined and 1 μg per sample was utilized for RNA-seq. Generation of sequencing libraries and sequencing were performed by Novogene Co Ltd (Cambridge, UK). Briefly, sequencing libraries were constructed using amplified cDNAs with the TruSeq Stranded Total RNA LT Sample Prep Kit (Illumina) and sequenced using the Illumina Novaseq6000 system in 150-bp paired-end mode. Bioinformatic analysis was performed by DREAMGENICS S.L. (Oviedo, Spain). First, sequenced reads were quality-checked with FastQC (http://www.bioinformatics.babraham.ac.uk/projects/fastqc/). The RNA reads were then aligned to the reference sequences of the Ensemble's 2019 version of the human genome (GRCh38) using the Salmon algorithm[60], and relative gene expression was quantified as transcript per million (TPM). For the analysis of differences in gene expression between conditions, the DESeq2 algorithm[61] was used. Differentially expressed genes were used as input for MSigDB GSEA[34]. Heatmap representation of differential gene expression for selected genes was performed using R Software (R version 4.1.0) and the "gplots" package. Gene expression level (z-score)

in experimental samples is represented as a colored cell, with a color key interpretation indicated where appropriate. Finally, an enrichment study by pathways and gene ontologies terms was performed using the PathfindR tool[62] and the repositories of Kyoto Encyclopedia of Genes and Genomes (KEGG) and Gene Ontology (GO).

## Protein extraction and Western blot (WB)

Cells were scraped off in RIPA buffer (0.1% SDS, 0.5% sodium deoxycholate, 1% NP40, 150 mM NaCl, 50 mM Tris–HCl pH 8.0, in PBS) supplemented with protease and phosphatase inhibitors (Sigma). 25 µg of total protein were boiled for 5′ at 95 °C, resolved by SDS-PAGE, and transferred to PVDF membranes in a Trans-Blot® SD Semi-Dry Transfer Cell (Bio-Rad). Unspecific binding to membranes was prevented by blocking in 5% bovine serum albumin (BSA) for 1 h at RT. For CXCR4 detection, a similar protocol was followed, but cell lysis was performed in 0.5% DDM buffer (0.5% n-dodecyl-b-d-maltoside, 140 mM NaCl, 25 mM Tris-HCl pH 7.4, 2 mM EDTA, in dH2O). In this case, 40 µg of total protein were loaded into the gel, samples were heated at 55 °C for 10′ and blocking was performed with milk. Primary antibodies [FAAH (Abcam #ab128917, dilution 1:1000), CXCR4 (Abcam #124824, dilution 1:500) or β-Actin (Sigma-Aldrich #A5441, dilution 1:5000)] in the blocking solution were incubated overnight (O/N) at 4 °C. Membranes were then washed three times with TBS-T, incubated for 1 h at room temperature (RT) with the corresponding HRP-conjugated secondary antibody (Cytiva Lifescience) and developed with self-prepared ECL reagent (1.25 mM luminol, 0.2 mM p-coumaric acid, 100 mM Tris–HCl pH 8.5, in dH2O). Densitometric analysis was performed with ImageJ™ open-source software version 1.52 (NIH)[63].

## Flow cytometry

Adherent cell lines were harvested using TrypLE™ Express (Gibco), washed with FACS buffer (PBS + 1% FBS + 1% BSA + 0.02% sodium azide) and incubated with hCXCR4 primary antibody (R&D Systems #MAB173, dilution 1:20) and then Alexa Fluor™ 647 Mouse IgG (Invitrogen #A-21235, dilution 1:200). 7-aminoactinomycin D (Biolegend, dilution 1:100) was used as a cell death marker and added 15′ before the analysis. In the case of breast tumors derived from MMTV-neu mice, a single cell solution was obtained from tumors after mechanical disaggregation followed by 2 h of enzymatic digestion in DMEM + 125 µg/mL collagenase (Sigma-Aldrich) at 37 °C. CXCR4 staining was performed as described above with already conjugated primary antibody Alexa Fluor™ 647 mCXCR4 (Biolegend #146503, dilution 1:100). Cell suspension was also incubated with biotinylated CD31 (Thermo Fisher Scientific #13-0311-82, dilution 1:200), CD45 (Thermo Fisher Scientific #13-0451-82, dilution 1:400) and TER119 (Thermo Fisher Scientific #13-5921-82, dilution 1:100) + PE Streptavidin (Biolegend #405203, dilution 1:300) to exclude endothelial and hematopoietic lineages (Lin+ cells) from the analysis. Data acquisition was performed in a FACSCalibur system (Becton Dickinson). For data analysis, FlowJo™ software version 10.6.2 (Becton, Dickinson and Company, 2019) was used. Fluorescence minus one controls were used to set the blank for CXCR4 expression after exclusion of cell debris, nonviable cells, and (in the case of MMTV-neu-derived tumors) Lin+ cells. A detailed guide to the gating strategy can be found in Supplementary Figs. 12 and 13.

## Immunohistochemistry (IHC)

Paraffin-embedded tissue sections were de-waxed in isoparaffin H (Panreac) and hydrated by quick changes through serial ethanol baths. Antigen retrieval was performed by boiling in sodium citrate buffer (10 mM sodium citrate, 0.05% Tween 20, pH 6.0, in dH2O). Samples were subjected to several washes with dH2O and washing buffer (NaCl 8.3 g/L, Tris 1.2 g/L; pH 7.4). Endogenous peroxidase activity was saturated by incubation with 3% H2O2. Blocking, primary [FAAH (Abcam #ab128917, dilution 1:100), PCNA (EMD Millipore, clone PC10, dilution 1:10.000) and CXCR4 (Abcam #124824, dilution 1:500)] and secondary antibody incubations and immunodetection were performed with the ImmPRESS™ Polymer Detection kit (Vector) following the manufacturer's instructions. Samples were finally counterstained with hematoxylin, dehydrated in ethanol, and mounted in Eukitt® Quick-hardening mounting medium (Sigma-Aldrich).

## Immunofluorescence and phalloidin staining

For the study of the EMT phenotype in BC cells, they were seeded on glass coverslips and fixed the day after in 10% formalin. Blocking was performed with 5% goat serum in PBS + 0.25% triton-X100 (Sigma-Aldrich) at RT. Primary antibodies [E-cadherin (Cell Signaling Technology #3195, dilution 1:200) and vimentin (BD Biosciences #550513, dilution 1:200)] in Antibody Diluent (Dako) were incubated O/N at 4 °C. Cells were then incubated with the corresponding fluorophore-conjugated secondary antibodies (Invitrogen) together with DAPI (Roche Life Science). For visualization of the actin filaments, Alexa Fluor™ 647 Phalloidin (Thermo Fisher Scientific #A22287, dilution 1:40 in 1% BSA) was added to the fixed cells. For immunofluorescence of MMTV-neu-derived tumors, paraffin-embedded tissue sections were de-waxed, hydrated, and boiled in sodium citrate buffer as described in the previous section. Blocking was performed with 10% goat serum in PBS. Primary antibodies [CK8 (DSHB #531826, dilution 1:50) and CK14 (Abcam #ab181595, dilution 1:50)] were diluted in Antibody Diluent and incubated O/N at 4 °C. Samples were then incubated with fluorophore-conjugated secondary antibodies and DAPI. Finally, they were mounted with Mowiol® mounting medium (Calbiochem). Fluorescence confocal images were acquired by using an Olympus FV1200 microscope and analyzed with ImageJ™ open-source software version 1.52 (NIH)[63].

## Cell invasion assays

Invasion assays for T-47D cells were performed using polyethylene terephthalate membranes manually coated with 3 mg/mL growth factor reduced (GFR) Matrigel® (Corning), while for MDA-MB-231 cells, BioCoat Matrigel® invasion chambers (Corning) were used. Briefly, cells were serum-starved O/N and 50 × 10³ were seeded in serum-free medium onto the upper compartment of the transwell while the lower compartment was filled with complete medium, 10% FBS acting as chemoattractant. After 48 h (MDA-MB-231) or 72 h (T-47D) of incubation, noninvasive cells were removed with a cotton swab. The cells that invaded through the membrane and adhered to the lower surface of the membrane were fixed with 10% formalin for 10′ and stained with DAPI. Cell invasion was quantified by counting cell nuclei in a fluorescence microscope.

## Mammosphere assay

Mammosphere medium was prepared by supplementing phenol red-free DMEM/F12 (Gibco) with 1X B27 minus vitamin A (Gibco), 20 ng/mL EGF (Gibco), 20 ng/mL FGF (Gibco), 2 µg/mL heparin (Sigma-Aldrich), 10 µg/mL insulin, 1 µg/mL hydrocortisone (Sigma-Aldrich), and 1% penicillin/streptomycin. T-47D cells were plated as a single-cell solution at a density of 500 cells/well in ultra-low attachment 6-well plates (Corning). After 5 days in culture, light microscopy at 40X magnification was used to assess mammosphere (>40 µm) formation. In our hands, the MDA-MB-231 cell line did not form mammospheres.

## Statistical analyses

Pearson's chi-squared test was used for statistical analysis of the human samples included in the TMAs and of the metastatic damage in lungs from mice injected with lung-seeking MDA-MB-231 cells. Kaplan-Meier survival curves were statistically compared by the log-rank test. Correlative associations between two variables were calculated by Pearson's $r$. Unpaired, independent groups of two were analyzed by two-tailed Student's t-test. For multi-group comparison, data were analyzed by 1-way ANOVA with post-Tukey's multiple comparison test, or by 2-way-ANOVA when required. A $p$ value of less than 0.05 was

considered statistically significant. Unless otherwise stated, data are expressed as mean ± SEM from at least three biological replicates. Statistical analysis of the collected numerical data was routinely performed with IBM SPSS Statistics version 28.0.1.1 and GraphPad Prism version 8.0.1.

## Reporting summary
Further information on research design is available in the Nature Portfolio Reporting Summary linked to this article.

## Data availability
All the raw data generated or analyzed during the study are included in this article, its Supplementary Information, and its Source Data file. In addition, the datasets generated for the RNA-seq of FAAH-modulated T-47D cells are available through the Gene Expression Omnibus (https://www.ncbi.nlm.nih.gov/geo/) under the accession number GSE197984. Source data are provided with this paper.

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

## Acknowledgements

This study has been funded by Instituto de Salud Carlos III (ISCIII) through the project PI17/00041 and PI20/00590 to C.San. and E.P.-G. and co-funded by the European Union. I.T. is the recipient of a PFIS fellowship (from the Spanish Ministry of Economy and Competitiveness). We are indebted to Eva Resel for administrative support. We also want to thank the sample donors, and the Biobank Hospital Universitario Puerta de Hierro, Majadahonda (HUPHM)/Instituto de Investigación Sanitaria Puerta de Hierro-Segovia de Arana (IDIPHISA) (PT17/0015/0020 in the Spanish National Biobanks Network) for the human specimens used in this study.

## Author contributions

I.T., C.San., and E.P.-G. designed the research and wrote the manuscript. I.T., M.S-V., S.B-B., M.R-H., S.A., C.A., S.M., I.H-M., and MM.C. performed the biological experiments. I.T., E.P-G., C.Sar., and M.V-M. performed computational and statistical analyses. I.T., S.B-B., M.G., C.San., MM.C., and E.P-G. supervised the research and reviewed and revised the manuscript. I.A-L., C.G-L., E.R-M., B.A., AJ.S-L., L.B., S.H., N.A., C.R., D.S., and G.M-B. provided technical and material support.

## Competing interests

The authors declare no competing interests.

## Additional information

[1]Department of Biochemistry and Molecular Biology, Complutense University, Madrid, Spain. [2]Instituto de Investigación Hospital 12 de Octubre, Madrid, Spain. [3]Brain Tumor Research Program, Telethon Kids Institute, Nedlands, WA, Australia. [4]Centre for Child Health Research, University of Western Australia, Nedlands, WA, Australia. [5]Breast Cancer Group, Oncology Area, Biodonostia Health Research Institute, San Sebastián, Spain. [6]Gipuzkoa Cancer Unit, OSI Donostialdea—Onkologikoa Foundation, San Sebastián, Spain. [7]Unit of Information and Healthcare Results, OSI Donostialdea, Biodonostia Health Research Institute, San Sebastián, Spain. [8]Methodological Support Unit, Biodonostia Health Research Institute, San Sebastián, Spain. [9]Centro de Biología Molecular Severo Ochoa (CBMSO) (CSIC-UAM), Madrid, Spain. [10]Department of Biology, Autonomous University of Madrid, Madrid, Spain. [11]Department of Pathology, Hospital Universitario Puerta de Hierro, Majadahonda, Madrid, Spain. [12]Department of Obstetrics & Gynecology, Hospital Universitario Puerta de Hierro, Majadahonda, Madrid, Spain. [13]Biobank Hospital Universitario Puerta de Hierro Majadahonda, Madrid, Spain. [14]Instituto de Investigación Sanitaria Puerta de Hierro-Segovia de Arana (IDIPHISA), Madrid, Spain. [15]Clinical Lipidomics Unit, Institute of Physiological Chemistry, University Medical Center, Mainz, Germany. [16]Department of Gynecology and Obstetrics, University Hospital Schleswig-Holstein, Kiel, Germany. [17]Institute of Pathology, University Hospital Schleswig-Holstein, Kiel, Germany. [18]Immunobiology Laboratory, Centro Nacional de Investigaciones Cardiovasculares (CNIC), Madrid, Spain. [19]MD Anderson International Foundation; Instituto de Investigaciones Biomédicas Alberto Sols (CSIC-UAM); Department of Biochemistry, Autonomous University of Madrid; Instituto de Investigación Hospital Universitario La Paz (IdiPaz); Centro de Investigación Biomédica en Red de Cáncer (CIBERONC), Madrid, Spain. [20]Ikerbasque—Basque Foundation for Science, Bilbao, Spain. [21]Instituto Ramón y Cajal de Investigación Sanitaria y Centro de Investigación Biomédica en Red de Enfermedades Neurodegenerativas (CIBERNED), Madrid, Spain. ✉e-mail: macsanch@ucm.es; eduperez@ucm.es

