## [Peer Review File · Nature Communications]

REVIEWER COMMENTS

Reviewer #1 (Remarks to the Author):

In this paper, the authors reported for the first time FAAH function as a new suppressor in regulating metastasis of breast cancer via decreasing AEA and CXCR4/CXCL12 signaling. Substantial data highly supported the main hypothesis, and of note, a lot of clinical analysis of both public data and clinical samples is impressive. Functional experiments on cell models were also solid. For further improvement of this study, concerns below should be noticed.

1. FAAH is of high expression in luminal breast cancers, and as shown in Fig. S2, its mRNA or protein level could be regulated by ER, likely acting as its TF. Thus, the analysis of expression correlation between FAAH and ESR should be taken into account, either with TGCA or bc-GenExMiner data.
2. Low FAAH indicates unfavorable outcome based on TAM#1 samples and KM database, which contain all subtypes BC patients. Specifically, TNBC with lowest FAAH expression (Fig. 1), is generally admitted as a subtype with poor prognosis and high rate metastasis. Additional analysis with only Luminal BCs and TAM#2 could be helpful.
3. Given that an inverse correlation between FAAH and HER2/TNBC status was stated, interestingly FAAH expression was linked to BC lung metastasis, but not bone or brain metastasis which HER2/TNBC subtypes preferred. Those clues suggested a particular role of FAAH in regulating metastasis of Luminal BCs, just as the statement in abstract and discussion which hints that FAAH has a specific role in Luminal BCs. However, it is also confused that FAAH similarly functions (via same downstream regulations) both in ER+ T-47D cells and TNBC MDA-MB-231 cells.
4. Again, whether Neu+(HER2) mice model is appropriate here needs more explanation (It should be noted that HER2 over-expression might disturb endogenous FAAH.), and the FAAH^{-/-} is whole body KO model?
5. In function experiments, eg. Fig. 4 c,e, s7e, and 6e, f, significance analysis between KO and rescue is absent.
6. In Fig. 6 and corresponding text, SDF1 and Cxcl12 should be used consistently. The inhibition of invasion effect of FAAH in Fig. 4e (231 cells) seems lost in Fig. 6f (the first column vs. the fourth column; the third column vs. the sixth column).
7. AEA is stated as a main mediator of FAAH function. If AEA influences cells' invasion, migration, or stemness in *in vitro* models?
8. In Fig. 7, AEA-induced CXCR4 expression could be partly reversed by SR1/SR2. However, whether high AEA promotes BC cells' metastasis *in vivo* and could be inhibited by CXCR4/cxcl12 inhibitors/antibodies is unknown. Thus further *in vivo* model could be important for the clinical significance of this study.
9. Additional discussion: Expression of FAAH is lost in metastatic tumors, does this due to ER decrease? Clinically, ER-targeting treatment could result in ER blocking or degradation; from this, FAAH should be down-regulated and increase the probability of cancer cell metastasis. Does this match the actual clinical data?

Reviewer #2 (Remarks to the Author):

In this work, the authors demonstrate that FAAH expression in BC is associated with luminal BC, and that patients with low tumor FAAH levels show significantly worse overall survival. They also show that the expression of FAAH in BC cells promotes a shift of the tumor phenotype towards a more differentiated state, reduces tumor progression and metastases in cellular and mouse models of BC. Finally, they demonstrate that these effects derive from the reduction of cannabinoid tone sustained mainly by Anandamide on CB1 and CB2 receptors. Taken together, these results highlight the role of FAAH as a metastasis suppressor in BC and as a new prognostic biomarker.

The work is very valid and with scientific rigor. The methods are well described, the data are clearly presented and their interpretation is correct and the conclusions are robust. In my opinion this work leads to a small but significant advancement for the research field.

In my opinion the work can be accepted in its current form.

Reviewer #3 (Remarks to the Author):

The authors make a strong case for the enzyme FAAH being important for the biology of breast cancer since they show in a genetic mouse model and in xenograft models that knocking out the enzyme interferes with disease manifestation.

Specific remarks.

1. The fact that FAAH is a valuable prognostic marker such be toned down. While the authors show that the effect observed is significant it will not likely be practice changing and does not add to established and clinically applied prognostic gene expression signatures (Mammaprint or Oncotype DX). Unless added value of this marker is show the finding may have clinical value for patients.
2. Details of the cases and their characteristics on TMA not described. How representative is the cohort used. Follow-up time is not mentioned, etc. Difficult to judge the findings without this knowledge. Also (and related to point 1) no multivariate analysis is performed.
3. The significance value of ER staining and FAAH staining is hard to read (increase the resolution or the size). Furthermore check the quality of the figures in general as they are hard or impossible to read (e.g. Fig 2e and suppl Fig 9b).
4. The statement that "84 % of all ER+ samples expressed detectable levels of FAAH (scores 1 to 3), while 85 % of all the ER- samples expressed no or low FAAH (scores 0 and 1, respectively)" (first paragraph of page 8) is misleading as 50% of the cases as score 1. Either report percentages for score 2 to 3 for ER+ or only score 0 for ER-.
5. On page 10 it is stated that "low tumor FAAH expression was linked to BC lung metastasis but not bone or brain metastasis (Supplementary Figs. 3d-f)". This is interesting but has this been corrected for subtype as these also tend to show differential preference for site of relapse? Anyhow to conclude this an interaction test needs to be done.
6. TMA2 is not ideal for looking at prognosis as this cohort only contains cases with matched lymph nodes. Thus lymph node negative cases which have a better prognosis are left from the equation. In relation to this Can the authors not analyse only the ER-positive cases from TMA1 to provide further support for the role of FAAH in luminal BC.
7. Considering the finding in TMA2 (i.e. drop in FAAH in lymph node metastasis) I wonder from a biological view why FAAH would drop in lymph node metastasis. Is this mainly ER driven? Thus the question arises is the disease in the lymph nodes still ER positive? Some rationale would help or are we not looking at some sort of bias? The link with the invasive front was made. Do the cells at the invasive front have less ER or a more stem cell/EMT or basal like features? It would be nice to confirm the markers from the mouse model (Figure 3) or the cell line model (supplemental Figure S7) in the clinical specimen, which would tie things better together.
8. In the mouse model EMT and adverse prognostic markers such Twist1, Snai2, Adam23, Abcb1a, Id1 are mentioned as significant but in fact they do not reach the uncorrected p value of 0.05. So I think this needs to be rephrased. The fact that no FDR was performed on the 84 genes analyzed further weakens this observation. As a group there might be an effect but as it is now there is none. Anyhow it is unclear how the 84 genes were selected? Are these prognostic factors, subtype related factors or what was the basis from them being included in the analysis. Are any of the markers negatively correlated to FAAH loss also in clinical specimen showing this behavior. This would also strengthen the finding.
9. With regard to the display of the 491 differentially expressed genes in Figure 1a, I find the presentation odd. The 491 genes seem now to relate to the parental (SCR) and the rescued lines (RESCUE). I assume these 2 lines show much fewer differentially expressed genes. Actually this

number preferably should be none but is not provided. It would be better to show the differentially expressed genes between the parental line and KO line in the one intersection and differential genes between the rescued and the KO line in the other and then indicate the 491 genes in the overlap which I expect to be large else I have serious concerns about the current finding.

10. Are the 84 genes studied in the mouse model above and the 76 genes studied the MDA231 derived metastatic lesions (Fig, 5e) detectable by RNAseq and do they behave in line with the in vitro data? With regard to consistency in figure 5 a 76 gene panel is used. It does not become clear why here a 76 "prognostic" gene set is used while elsewhere a 84 gene set.

11. I am not too convinced by CXCR4 protein staining, particularly in the MDA-MB-231 sublines. It is unclear what is a specific signal. Shift of the curves in relation to mock-staining is not shown.

12. An significant inverse association between CXCR4 and FAAH expression was noted suggesting causality. My team and others have shown that CXCR4 is more expressed in less luminal tumors so the link of both markers with ER remains a confounding factor. The same hold true for the CXCR4 pathway studied in suppl figure S9e in the Wang et al dataset (why only the Wang dataset is analysed here also does not become clear). Is the relation still there if ER status is taken into consideration as a confounder.

13. Combining CXCR4 with FAAH to show prognostic value is misleading since FAAH in itself is already a marker providing the levels of significance observed here. It would be more fair to at least also show that CXCR4 on its own has significance since both markers are considered to have a shared relation to the measured end-point. Therefore combining them will not learn us anything. More informative would be to see how CXCR4 levels behave in the tumors that show a difference between the primary tumor and the lymph node metastasis for FAAH and those that do not.

14. Unfortunately it do not become clear how FAAH is directly connected to CXCR4. Or at least it does not become clear of the effect of metAEA indeed phenocopied the effect of FAAH via the same pathway. Transcriptome or marker analyses should be done to show the same pathway are used. I am also concerned about the involvement of bonified cannabis receptor as gene expression of CNR and CNR2 is undetectable in BC tumors and cell lines (including the one used here). So I would appreciate the effect seen here is meditated by one of the cannabis receptor directly (e.g. by using knock-outs).

RESPONSE TO REVIEWERS' COMMENTS

POINT-BY-POINT ANSWER TO REVIEWER #1

We would like to thank Reviewer #1 for his/her constructive comments and suggestions, that led us to conduct additional analyses and experiments that, in our opinion, have significantly improved the manuscript and strengthened its conclusions. Below are the answers to all his/her comments:

“In this paper, the authors reported for the first time FAAH function as a new suppressor in regulating metastasis of breast cancer via decreasing AEA and CXCR4/CXCL12 signaling. Substantial data highly supported the main hypothesis, and of note, a lot of clinical analysis of both public data and clinical samples is impressive. Functional experiments on cell models were also solid. For further improvement of this study, concerns below should be noticed.”

We thank the reviewer for his/her positive comments on our manuscript.

“1.FAAH is of high expression in luminal breast cancers, and as shown in Fig. S2, its mRNA or protein level could be regulated by ER, likely acting as its TF. Thus, the analysis of expression correlation between FAAH and ESR should be take into account, either with TGCA or bc-GenExMiner data.”

As suggested by the reviewer, we performed a correlation analysis between *ESR1* and *FAAH* expression in both the TGCA and the bc-GenExMiner datasets. Moreover, we run a similar analysis in two additional public datasets (METABRIC and Neve *et al.*, 2006) including transcriptomic data from human tumors and BC cell lines. The results of all these analyses, which are shown in Fig. R1a-d in the file entitled “Figures for Reviewers”, demonstrate a strong positive correlation between the expression of the two genes in BC. We have not included these new results in the manuscript but will be happy to do it if the reviewer considers that it is necessary.

“2.Low FAAH indicates unfavorable outcome based on TAM#1 samples and KM database, which contain all subtypes BC patients. Specifically, TNBC with lowest FAAH expression (Fig. 1), is generally admitted as a subtype with poor prognosis and high rate metastasis. Additional analysis with only Luminal BCs and TAM#2 could be help.”

As the reviewer very rightly points out, TMA #1 contains tumors from all BC molecular subtypes (Fig. 1c, d), and therefore the relation between high FAAH and a better outcome (Fig. 2a, b) may be biased by the fact that FAAH is mostly expressed in luminal tumors, which are inherently related to a better prognosis. This is in fact contemplated in the manuscript, and it is the reason why TMA #2 (made up by luminal tumors only) was included in the study:

“Prognostic associations inferred from tumor data including all intrinsic BC subtypes may be biased considering that FAAH has a predominant expression in the luminal subtype of BC, which is the one with the highest inherent survival rates. To rule out the possibility that our conclusions were affected by this limitation, we next analyzed FAAH expression in a second TMA (i.e., TMA #2) containing primary non-treated breast tumor samples exclusively of the luminal subtype (n=276) and matched synchronous lymph node (LN) metastatic samples.” Results section, page 9.

In TMA #2, a strong association between low FAAH expression and poor outcome was found both in the primary tumor (Supplementary Fig. 5b) and in the lymph node (Supplementary Fig. 5c), thus demonstrating

that FAAH expression is related to patient prognosis regardless of its association with the luminal BC subtype.

In order to strengthen these observations, and as specifically requested by the reviewer, we performed the association analysis on TMA #1 with luminal BC tumor samples only, and found again that low FAAH expression levels are associated with a poor patient outcome (Fig. R2).

“3. Given that an inverse correlation between FAAH and HER2/TNBC status was stated, interestingly FAAH expression was linked to BC lung metastasis, but not bone or brain metastasis which HER2/TNBC subtypes preferred. Those clues suggested a particular role of FAAH in regulating metastasis of Luminal BCs, just as the statement in abstract and discussion which hints that FAAH has specific role in Luminal BCs. However, it is also confused that FAAH similarly function (via same downstream regulations) both in ER+ T-47D cells and TNBC MDA-MB-231 cells.”

We apologize if the conclusions from this part of the manuscript were not clear enough. In this study, we demonstrate that FAAH is a marker of luminal BC, and that its high expression in this subtype is associated with a better patient outcome. However, the role of FAAH in regulating tumorigenic traits is not exclusive of the luminal subtype. The experiments performed in this manuscript reveal that FAAH activity is necessary to maintain (in the case of luminal BC) or induce (in the case of basal-like BC) luminal-like phenotypes, and that -in both cases- it does so by regulating the bioavailability of endogenous anandamide at cannabinoid receptors and, at least in part, the subsequent modulation of the CXCR4 signaling pathway. Under this rationale, tumors with low FAAH expression (regardless of their subtype) would tend to present a basal-like genetic signature and, therefore, to develop organ-specific metastases to those sites where basal-like tumors most frequently metastasize, such as the lung (Kennecke *et al.*, 2010). Consistent with this notion, it was not surprising to find that lower FAAH expression was associated with lung metastasis in the datasets depicted in Supplementary Fig. 3. We hope that this explanation clarifies the reviewer's doubts.

“4. Again, whether Neu+(HER2) mice model is appropriate here needs more explanation (It should be noted that HER2 over-expression might disturb endogenous FAAH.), and the FAAH^{-/-} is whole body KO model?”

In this work, we demonstrate that FAAH expression in human samples is associated with the luminal BC subtype and lung metastasis, and that its genetic silencing in luminal BC cells in culture switches them into overt basal-like phenotypes. To validate the significance of these observations *in vivo* and further strengthen cause-and-effect links, we needed an animal model that developed (1) luminal-like breast tumors and (2) lung metastases, and in which (3) we could knock-out FAAH (by crossing with FAAH^{-/-} mice). The MMTV-neu mouse fulfills these requirements as it generates tumors and lung metastasis that, despite being driven by the *Neu* oncogene, have a global gene expression profile more similar to luminal A and B than to HER2+ subtypes:

- **Identification of conserved gene expression features between murine mammary carcinoma models and human breast tumors.** Herschkowitz *et al.* Genome Biol 2007.

“*TgMMTV-Neu* tumors did not have a significant gene overlap with the human HER2+/ER- subtype and were more similar to human luminal tumors.”

- **Transcriptome analyses of mouse and human mammary cell subpopulations reveal multiple conserved genes and pathways.** Lim *et al.* Breast Cancer Res 2010.

“The MMTV-Neu strain, however, does not accurately recapitulate HER2-overexpressing cancers arising in women, because MMTV-Neu tumors do not show significant gene overlap with the HER2-positive subtype but are more similar to human luminal tumors.”

- **Transcriptomic classification of genetically engineered mouse models of breast cancer identifies human subtype counterparts.** Pfefferle *et al.* Genome Biol 2013.

“Previously defined as a ‘luminal’ model, the Neu^{Ex} murine class associated with the human luminal A subtype in this newest analysis.”

In addition, to the best of our knowledge, there is no reported connection between the *Neu* oncogene and FAAH that could render the MMTV-neu model inadequate for studying the role of FAAH in tumorigenesis, nor any evidence that *Neu* is able to regulate the endogenous levels of FAAH in BC. For all these reasons, we firmly believe that the MMTV-neu model fits the needs of our work and can provide valuable and unbiased information on the role of FAAH in BC. We also confirm that the FAAH^{-/-} mice used were full-body KO (Cravatt *et al.*, 2001).

“5.In function experiments, eg. Fig.4 c,e, s7e, and 6e, f, significance analysis between KO and rescue is absent.”

We thank the reviewer for the observations. We have addressed this issue as follows:

- In Figs. 4c (endocannabinoid levels) and 4e (invasion assay), we performed additional experiments to increase statistical power and the ANOVA with post Tukey’s multiple comparison test was repeated, rendering a p value < 0.05. The corresponding figures were modified accordingly.
- In Supplementary Fig. 7e (mammosphere formation), we detected that an inadequate statistical test had been used (Student’s t-test instead of ANOVA). We sincerely apologize for this. We performed 1-way ANOVA with post Tukey’s multiple comparison test, rendering a p value < 0.05. The corresponding figure was modified accordingly.
- In current Fig. 6d (former Fig. 6e: invasion assay in the presence of AMD3100), we performed additional experiments to increase statistical power. In this case, and most probably due to the considerable inter-experimental variability inherent to the invasion assays, no statistical significance between KO and rescue groups was reached in the ANOVA test.

“6.In Fig. 6 and corresponding text, SDF1 and Cxcl12 should used consistently.”

We apologize for the interchangeable use of SDF1 and CXCL12. SDF1 has been replaced by CXCL12 throughout the text and figures.

“The inhibition of invasion effect of FAAH in Fig. 4e (231 cells) seems lost in Fig.6f (the first column vs. the fourth column; the third column vs. the sixth column).”

We apologize for the misunderstanding the different experimental conditions of the invasion assays in Figs. 4 and 6 may have caused. In Fig. 4e, we are evaluating the invasion capacity of MDA-MB-231 cells in response to the most commonly used chemoattractant in invasion assays, which is 10 % FBS (this is stated in the figure legend). In current Fig. 6e (former Fig. 6f), however, we are evaluating the ability of MDA-MB-231 cells to invade towards CXCL12, so chemoattractant in this case is CXCL12 only (this is stated in the figure legend as well). Because the lower invasion capacity of FAAH-overexpressing MDA-MB-231 cells probably lies in many factors apart from their lower CXCR4 levels, differences in invasion between parental and FAAH-overexpressing MDA-MB-231 cells are expected to be greater when the chemoattractant includes many other chemokines and growth factors FAAH-overexpressing cells might be less attracted to, such as the case of 10 % FBS.

“7.AEA is stated as a main mediator of FAAH function. If AEA influence cells' invasion, migration, or stemless on in vitro models?”

In this work, we show that FAAH regulates the tumorigenic traits of BC cells through the modulation of the endogenous levels of AEA and its subsequent action at cannabinoid receptors (CBRs). The original version of the manuscript included some experiments showing that metAEA mimics the effect of FAAH genetic silencing in T-47D cells [induction of CXCR4 expression (Fig. 7a) and increase in invasion both in T-47D and MDA-MB-231 cells (Fig. 7b)]. However, we agree with the reviewer in the need of additional proofs for the involvement of AEA in FAAH-mediated actions. We have therefore performed additional experiments with metAEA, that have been included in this revised version of the manuscript, as follows:

- Previously, we had demonstrated that knocking-out FAAH in T-47D cells induced an EMT phenotype *in vitro* (Supplementary Fig. 7c), and that FAAH overexpression in MDA-MB-231 cells attenuated it (Supplementary Fig. 7d). In the **(new) Supplementary Fig. 10a**, we now demonstrate that metAEA treatment over a long period (4 weeks) also induces an EMT-like phenotype in T-47D cells through the activation of CBRs.
- Previously, we had demonstrated that lung-tropic MDA-MB-231 cells overexpressing FAAH formed less lung metastasis than parental cells (Fig. 4i). To demonstrate that this phenotype was due to their lower endogenous AEA levels, we have performed a first *in vivo* experiment in which mice were injected with FAAH-overexpressing lung-tropic MDA-MB-231 cells and subsequently treated with different doses of metAEA (0.1, 1 and 10 mg/kg). **Fig. R3** shows that metAEA, as parental cells with low levels of FAAH, increased the metastatic burden of these animals. Once the pro-metastatic effect of metAEA was demonstrated, we performed a second *in vivo* experiment to try to demonstrate the involvement of CBRs and CXCR4 in metAEA action. The results of this experiment have been included in the revised version of the manuscript as **(new) Fig. 7d**, and show that the pro-metastatic effect of metAEA is prevented by CB₁R, CB₂R and CXCR4 selective antagonists, further reinforcing the participation of CBRs and CXCR4 in FAAH-driven metAEA-mediated actions.
- Previously, we had demonstrated that metAEA increased the invasive capacity of T-47D and MDA-MB-231 cells (Fig. 7b), and that this effect was prevented in T-47D cells by blocking CB₁R, CB₂R, and CXCR4 with selective antagonists. We have now included these conditions (SR1, SR2 and

AMD3100) in MDA-MB-231 cells and have obtained similar results, which are included in the revised manuscript as **Fig. 7c**.

The revised text has been modified accordingly to describe and comment these new experiments both in the Results and Discussion sections, respectively.

We strongly believe that these new findings add a solid proof for the FAAH/AEA/CBRs/CXCR4 mechanistic connection in BC, which was, as pointed out by two of the three reviewers (and we honestly agree), the weakest part of our original work.

“8. In Fig. 7, AEA-induced CXCR4 expression could be partly reversed by SR1/SR2. However, whether high AEA promotes BC cells' metastasis *in vivo* and could be inhibited by CXCR4/cxcl12 inhibitors/antibodies is unknown. Thus further *in vivo* model could be important for the clinical significance of this study.”

The new *in vitro* and *in vivo* experiments carried out to strengthen the mechanistic connection between FAAH, AEA, CBR2 and CXCR4 in BC are described in detail in the previous point.

“9. Additional discussion: Expression of FAAH is lost in metastatic tumors, does this due to ER decrease?”

In this study, we demonstrated that FAAH and ER expression positively correlate in human BC samples, and that, in BC cell lines, FAAH expression is induced by estrogens and inhibited by knocking-out ER. Considering that one of the main survival-affecting events in luminal BC is the loss of ER dependence, resulting in resistance to ER-targeted therapies and, eventually, progression to metastatic disease (Lower *et al.*, 2005; Szostakowska *et al.*, 2019), it would be tempting to speculate that luminal BC metastases would have lower expression of ER and ER downstream targets such as FAAH. However, luminal BC metastases have been shown to contain both ER+ and ER- lesions, indicating a heterogeneous seeding or the selective loss of ER positivity in disseminated cells (Ogba *et al.*, 2014). In the tumor samples included in TMA #2, loss of ER expression is observed only in 5.75 % of the patients, and, within this subpopulation, 40 % also showed FAAH downregulation, 12 % upregulation, and 48 % no changes in FAAH expression (data not shown). Collectively, these observations support the accepted notion that reduced ER dependence of luminal BC metastases relies on many factors other than downregulation of ER, such as an altered expression of ER co-regulators or the presence of ER mutations, that ultimately lead to a blockade of ER signaling (Lopez-Knowles *et al.*, 2019; Martin *et al.*, 2017). Therefore, the existence of other upstream regulators of FAAH expression that may explain its downregulation in BC metastases cannot be ruled out.

“Clinically, ER-targeting treatment could result in ER blocking or degradation; from this, FAAH should be down-regulated and increase the probability of cancer cell metastasis. Does this match the actual clinical data?”

Because estrogen is a major activator of proliferation in luminal tumors, ER and its downstream signaling are excellent targets for endocrine therapy in luminal BC patients. More specifically, large-scale randomized trials have shown that, in early-stage ER+ breast cancers, a 5-year course of tamoxifen started immediately after surgery reduces recurrence by 51 % and mortality by 28 % (EBCTCG, 1998). However, a significant proportion of patients exhibit *de novo* or acquired resistance to endocrine therapy, and up to 20 % of patients subsequently relapse and die from metastatic disease after 5 years (Arimidex *et al.*, 2008). As the reviewer

points out, ER-targeting is expected to result in FAAH downregulation in tumor cells and/or in the clonal selection of low FAAH-expressing populations that could survive and eventually recur and form metastases. This suggests that the downregulation of FAAH following endocrine treatment could be involved in some of the resistance mechanisms to this type of therapy. Following this rationale, intrinsically low FAAH values in the tumor could work as a predictor of relapse themselves, given that (as discussed above) they are an indirect measure of a low ER-dependence even in ER+ cells. In order to validate these assumptions, we studied FAAH mRNA levels in a series of 132 primary tumors from patients who received adjuvant tamoxifen (Chanrion *et al.*, 2008) and found out that tumor samples with low FAAH levels had a significantly lower relapse-free survival (**Fig. R4**). These observations suggest that low FAAH expression in the tumor may serve to identify those patients at a higher risk of relapse after ER-targeting endocrine therapy, and make them prone to gain benefits from complementary/alternative treatments such as adjuvant chemotherapy.

POINT-BY-POINT ANSWER TO REVIEWER #2

“In this work, the authors demonstrate that FAAH expression in BC is associated with luminal BC, and that patients with low tumor FAAH levels show significantly worse overall survival. They also show that the expression of FAAH in BC cells promotes a shift of the tumor phenotype towards a more differentiated state, reduces tumor progression and metastases in cellular and mouse models of BC. Finally, they demonstrate that these effects derive from the reduction of cannabinoid tone sustained mainly by Anandamide on CB1 and CB2 receptors.

Taken together, these results highlight the role of FAAH as a metastasis suppressor in BC and as a new prognostic biomarker.

The work is very valid and with scientific rigor. The methods are well described, the data are clearly presented and their interpretation is correct and the conclusions are robust. In my opinion this work leads to a small but significant advancement for the research field.

In my opinion the work can be accepted in its current form.”

We thank the reviewer for his/her positive comments on our manuscript.

POINT-BY-POINT ANSWER TO REVIEWER #3

We would like to thank Reviewer #3 for his/her constructive comments and suggestions, that led us to conduct additional analyses and experiments that, in our opinion, have significantly improved the manuscript and strengthened its conclusions. Below are the answers to all his/her comments:

“The authors make a strong case for the enzyme FAAH being important for the biology of breast cancer since they show in a genetic mouse model and in xenograft models that knocking out the enzyme interferes with disease manifestation.

Specific remarks.

1. The fact that FAAH is a valuable prognostic marker such be toned down. While the authors show that the effect observed is significant it will not likely be practice changing and does not add to established and clinically applied prognostic gene expression signatures (Mammaprint or Oncotype DX). Unless added value of this marker is show the finding may have clinical value for patients.”

We apologize if the wording of our manuscript suggested that our results were going to be practice changing. That was never our intention. We have always tried to be extremely cautious with the interpretation of our results and the possible translational impact of our work. In this manuscript, FAAH expression is demonstrated to be significantly associated with clinical parameters such as overall survival and metastasis-free survival of the patients, and these associations are supported by statistical analyses. However, we are fully aware of the limitations of our findings, and therefore, any reference to the role of FAAH as a new biomarker is accompanied in the text by the words “potential” or “suggest”, or the conditional tense.

As requested by the reviewer, we have toned down our message by including a sentence in the Discussion section (page 20, second paragraph) stating that the actual translational potential of FAAH as a valuable prognostic marker would require, aside from validation in larger cohorts of patients, the demonstration that it performs better than the currently used prognostic platforms such as Mammaprint or Oncotype DX. We do hope that way we will satisfy the reviewer’s concern, but will be of course willing to consider additional or alternative options.

“2. Details of the cases and their characteristics on TMA not described. How representative is the cohort used. Follow-up time is not mentioned, etc. Difficult to judge the findings without this knowledge. Also (and related to point 1) no multivariate analysis is performed.”

We apologize for the lack of detailed information on the characteristics of the patient cohorts in TMA #1 and TMA #2 and follow-up time. The later (follow-up time) has been added to the Materials & Methods section, and the clinical, pathological and immunohistochemical information of the cases has been included in the revised version of the manuscript as **Supplementary Data 1**. In addition, we performed multivariate analysis with the data of TMA #1 (at the end of the file “Figures for Reviewers”) and the results indicate that FAAH expression is an independent prognostic factor.

“3. The significance value of ER staining and FAAH staining is hard to read (increase the resolution or the size). Furthermore check the quality of the figures in general as they are hard or impossible to read (e.g. Fig 2e and suppl Fig 9b).”

The original version of the manuscript (the one the reviewers have received) was a non-high-quality pdf, as per journal’s directions. If accepted, the manuscript will be published with figures in high resolution. To address the reviewer’s concern, bigger and higher quality pictures have been included in a file entitled “Figures for Reviewers”. Specifically, those containing FAAH and ER staining appear in **Figs. R5A and R5B**. In addition, the size of the text in all the figures of the manuscript has been revised and increased.

“4. The statement that “84 % of all ER+ samples expressed detectable levels of FAAH (scores 1 to 3), while 85 % of all the ER- samples expressed no or low FAAH (scores 0 and 1, respectively)” (first paragraph of page 8) is misleading as 50% of the cases as score 1. Either report percentages for score 2 to 3 for ER+ or only score 0 for ER-.”

We agree with the reviewer that the statement was misleading, and therefore we have removed it from the text.

“5. On page 10 it is stated that “low tumor FAAH expression was linked to BC lung metastasis but not bone or brain metastasis (Supplementary Figs. 3d-f)”. This is interesting but has this been corrected for subtype as these also tend to show differential preference for site of relapse? Anyhow to conclude this an interaction test needs to be done.”

The reviewer makes an excellent point related to the preferential metastatic tropism associated to the different BC subtypes. We agree with him/her that the only way to establish an unbiased, reliable relation between FAAH and a site-specific metastasis would be to perform an analysis segregated by BC subtypes. Unfortunately, the datasets we have used to study the metastatic tropism of tumors with high and low FAAH expression (van de Vijver *et al.*, 2002; Wang *et al.*, 2005) do not contain information regarding tumor subtype. Despite this, we believe that this figure hints valuable information. For example, the finding that FAAH high-expressing tumors are not preferentially associated to bone metastasis (which, given that FAAH is mostly expressed by luminal tumors, is the location where they would be expected to metastasize) suggests that these observations are not necessarily biased by tumor subtype. In addition, the role of FAAH in lung metastasis is supported by *in vivo* experiments included in this manuscript (*i.e.*, MMTV-neu mice model and lung-tropic MDA-MB-231 model). In spite of this, and as implied by the reviewer, further studies are warranted to unequivocally demonstrate the role of FAAH in organ-specific metastases.

“6. TMA2 is not ideal for looking at prognosis as this cohort only contains cases with matched lymph nodes. Thus lymph node negative cases which have a better prognosis are left from the equation. In relation to this Can the authors not analyse only the ER-positive cases from TMA1 to provide further support for the role of FAAH in luminal BC.”

As requested by the reviewer, we performed the association analysis on TMA #1 in the luminal tumor samples and found that low FAAH expression associates with poor patient outcome in this subset. Results are shown in **Fig. R2**.

“7. Condering the finding in TMA2 (i.e. drop in FAAH in lymph node metastasis) I wonder from a biological view why FAAH would drop in lymph node metastasis. Is this mainly ER driven? Thus the question arises is the disease in the lymph nodes still ER positive? Some rationale would help or are we not looking at same sort of bias?”

One of the main survival-affecting events in luminal BC is the loss of ER dependence, resulting in resistance to ER-targeted therapies and, eventually, progression to metastatic disease (Lower *et al.*, 2005; Szostakowska *et al.*, 2019). Interestingly, luminal BC metastases have been shown to contain both ER+ and ER- lesions, indicating heterogeneous seeding or the selective loss of ER positivity in disseminated cells (Ogba *et al.*, 2014). In the tumor series included in TMA #2, there is a loss of ER expression in only 5.75 % of the patients, and, within this subpopulation, 40 % also showed FAAH downregulation, 12 % upregulation, and 48 % no changes in FAAH expression (data not shown). However, there is evidence that reduced ER dependence of luminal BC metastases relies on many factors other than ER downregulation, such as an altered expression of ER co-regulators or the presence of ER mutations that ultimately lead to a blockade of ER signaling (Lopez-Knowles *et al.*, 2019; Martin *et al.*, 2017). In this work, we demonstrate that FAAH and ER expression positively correlate in human BC samples, and that, in BC cell lines, FAAH expression is induced by estrogens and inhibited by knocking-out ER. One would expect that, given that luminal BC metastases are less dependent on ER signaling (even if there is no downregulation of ER expression), they would also show a lower expression of ER targets such as FAAH. However, the existence of other upstream regulators of FAAH expression that may explain its downregulation in BC metastases cannot be ruled out.

“The link with the invasive front was made. Do the cells at the invasive front have less ER or a more stem cell/EMT or basal like features? It would be nice to confirm the markers from the mouse model (Figure 3) or the cell line model (supplemental Figure S7) in the clinical specimen, which would tie things better together.”

The reviewer raises a very interesting point. Some labs have described a process characterized by tumor cells at the invasive front showing different morphologies and preferentially expressing certain proteins compared with tumor cells at the center of the tumor mass (Ono *et al.*, 1996). Although this has been most extensively described in colorectal cancers, it has also been observed in BC in the so-called prairie fire pattern (Kauppila *et al.*, 1998). To better characterize the molecular features of the invasive-like structures shown in Supplementary Fig. 5d, we performed IHC staining of several luminal/basal and EMT markers on samples from the same patient (patient #42). These new data are included in **Fig. R6**, and show a downregulation of ER in some of the low FAAH-expressing cells at the invasion front as well as in some of those that have escaped the primary tumor. We also performed IHC of basal/mesenchymal markers (*i.e.*, CK14 and vimentin) but were not able to find any positive staining, most probably due to the luminal nature of the tumors included in TMA #2. The limited availability of human samples, as well as the difficulty to identify a tumor section in which the same invasive-like structures were found, made it challenging to perform further analyses.

“8. In the mouse model EMT and adverse prognostic markers such Twist1, Snai2, Adam23, Abcb1a, Id1 are mentioned as significant but in fact they do not reach the uncorrected p value of 0.05. So I think this needs to be rephrased.”

Fig. 3h and (current) Supplementary Data 2 (former Supplementary Data 1) show that tumors derived from MMTV-neu:FAAH^{-/-} mice express a gene signature characteristic of more undifferentiated, basal-like tumors. However, and as pointed out by the reviewer, some of the differences in expression (*Twist1*, *Snai2*, *Adam23*, *Abcb1a*, and *Id1*) did not reach p values < 0.05 and, therefore, are not considered statistically significant. This is most probably due to the small number of biological replicates (n=3 mice per genotype) and the considerable inter-experimental variability. In these cases, the observed differences as referred to as trends in the text (specifically, “expression-upregulation trend”) and not as statistically significant differences.

“The fact that no FDR was performed on the 84 genes analyzed further weakens this observation. As a group there might be an effect but as it is now there is none. Anyhow it is unclear how the 84 genes were selected? Are these prognostic factors, subtype related factors or what was the basis from them being included in the analysis. Are any of the markers negatively correlated to FAAH loss also in clinical specimen showing this behavior. This would also strengthen the finding.”

We apologize for the misunderstanding that the use of the RT² Profiler PCR Array System by Qiagen may have caused. These are commercial panels of qPCR assays consisting of 84 different gene probes related to specific signaling pathways, cell functions or diseases. In this study we used two different panels:

- **RT² Profiler™ PCR Array Mouse Breast Cancer (#PAMM-131Z)**. It includes BC-related genes belonging to the following categories (as established by the manufacturer): BC classification markers, signal transduction, EMT, angiogenesis, cell adhesion, proteases, apoptosis, DNA damage and repair, xenobiotic transport, and transcription factors. We used this kit to perform an initial screening of the BC-related functions that were different in tumors derived from MMTV-neu:FAAH^{+/+} and FAAH^{-/-} mice with the aim of understanding the role of FAAH in BC tumorigenesis.
- **RT² Profiler™ PCR Array Human Tumor Metastasis (#PAHS-028Z)**. It includes metastasis-related genes belonging to the following categories (as established by the manufacturer): cell adhesion molecules, extracellular matrix molecules, cell cycle, cell growth and proliferation, apoptosis, transcription factors and regulators and other tumor metastasis genes. Since we already had observed a functional connection between FAAH and the invasion potential of BC cells, we used this kit to study invasion and metastasis-related genes that were differentially expressed in FAAH-overexpressing MDA-MB-231 cells.

No FDR was performed because data representation and statistical analysis were carried out with the corresponding tool provided by the supplier (Qiagen), which calculates the fold change and the p values.

In the different transcriptomic studies we have performed to find FAAH-regulated molecular pathways that could explain its role in tumorigenesis (*i.e.*, RNA-seq and both of the PCR Arrays described above), *FAAH* expression was found to negatively associate with several markers related to basal-like features, stemness and pro-invasive and pro-metastatic phenotypes. As requested by the reviewer, we have studied the expression of some of these candidates in human BC samples from the METABRIC dataset and found that these negative associations occurred in clinical samples too, further strengthening our findings from the preclinical models. These new results are included in **Fig. R7A-C**.

“9. With regard to the display of the 491 differentially expressed genes in Figure 1a, I find the presentation odd. The 491 genes seem now to relate to the parental (SCR) and the rescued lines (RESCUE). I assume these 2 lines show much fewer differentially expressed genes. Actually this number preferably should be none but is not provided. It would be better to show the differentially expressed genes between the parental line and KO line in the one intersection and differential genes between the rescued and the KO line in the other and then indicate the 491 genes in the overlap which I expect to be large else I have serious concerns about the current finding.”

We are sorry that the Venn diagram we used for the representation of the RNA-seq results was so confusing. We have replaced it with a heatmap (Fig. 5a) showing the comparison of the gene expression between FAAH KO cells and both the SCR and the RESCUE groups. Just for clarification, differential expression analysis was performed by Dreamgenics S.L. (the same company that performed the RNA-seq) based on our biological question: “which are the genes that are differentially expressed in the FAAH KO vs. SCR cells that also recovered their original levels in the RESCUE subline?” In other words, we were interested in those genes whose expression overlapped in the SCR/RESCUE groups but did not overlap with the FAAH KO (these were 491).

“10. Are the 84 genes studied in the mouse model above and the 76 genes studied the MDA231 derived metastatic lesions (Fig, 5e) detectable by RNAseq and do they behave in line with the in vitro data?”

- Regarding the genes included in the RT² Profiler™ PCR Array Mouse Breast Cancer, 12 of them were detected as differentially expressed in T-47D FAAH KO cells and recovered their levels in rescue cells. Out of these, 10 (*THBS1*, *TFF3*, *ESR1*, *CDKN1A*, *VEGFA*, *KRT18*, *CDH1*, *TWIST1*, *CTNNB1* and *AR*) were found upregulated and 2 (*IGFR1*, *ID1*) downregulated in the FAAH KO cells. Interestingly, most of the upregulated ones (*TFF3*, *CDKN1A*, *VEGFA*, *CDH1*, *TWIST1*, *CTNNB1* and *AR*) have been described as protumorigenic genes or risk factors in BC, which is in line with the more aggressive behavior of T-47D FAAH KO cells. On the other hand, protumorigenic functions have also been attributed to *IGFR1* and *ID1*, which were downregulated in FAAH KO cells. Future studies might deepen into the relative contribution of each of these genes to the phenotype of FAAH KO cells and the protumorigenic actions driven by FAAH in BC.
- Regarding the genes included in the RT² Profiler™ PCR Array Human Tumor Metastasis, 10 of them were detected as differentially expressed in T-47D FAAH KO cells and recovered their levels in rescued cells. Out of these, 5 (*CXCR4*, *TIMP3*, *CD44*, *VEGFA* and *CXCL12*) were found upregulated and 5 (*HRAS*, *EPHB2*, *TP53*, *FGFR4* and *MTA1*) downregulated in the FAAH KO cells. Except for *TIMP3*, all upregulated genes are well-established risk factors in BC. On the other hand, both BC risk factors (*HRAS*, *EPHB2*, *MTA1*) and tumor suppressor genes (*TP53*) were found among the downregulated genes. Future studies might shed light on the relative contribution of each of these genes to the phenotype of FAAH KO cells and on the protumorigenic actions driven by FAAH in BC.

“With regard to consistency in figure 5 a 76 gene panel is used. It does not become clear why here a 76 “prognostic” gene set is used while elsewhere a 84 gene set.”

We apologize for this misunderstanding. As mentioned above, these are commercial panels of qPCR assays consisting of 84 different gene probes each. However, in the RT² Profiler™ PCR Array Human Tumor Metastasis we used for FAAH-modulated MDA-MB-231 cells, no expression was reported for 8 out of these 84 genes, and therefore it was described in the text as a panel with 76 genes only. We agree with the reviewer that this is confusing and makes the number of chosen genes seem arbitrary, so we have modified the manuscript accordingly, clarifying that these are always 84-gene panels.

“11. I am not too convinced by CXCR4 protein staining, particularly in the MDA-MB-231 sublines. It is unclear what is a specific signal. Shift of the curves in relation to mock-staining is not shown.”

In the flow cytometry experiments for CXCR4 detection, mock-staining was subtracted from the specific CXCR4 signal by using fluorescence-minus-one (FMO) controls. A detailed guide of the gating strategy has been now included as **Supplementary Material**, with one of the MDA-MB-231 samples in Fig. 6a being used as an example.

“12. An significant inverse association between CXCR4 and FAAH expression was noted suggesting causality. My team and others have shown that CXCR4 is more expressed in less luminal tumors so the link of both markers with ER remains a confounding factor.”

The reviewer raises the very interesting and accurate point that the inverse relation between *FAAH* and *CXCR4* could be biased by the preferential expression of *CXCR4* by ER- tumors. As shown in **Fig. R8**, we have performed a correlation analysis in ER+ and ER- BC samples of the METABRIC dataset and have verified that the inverse association between *FAAH* and *CXCR4* occurs in all cases, regardless of the ER status.

“The same hold true for the CXCR4 pathway studied in suppl figure S9e in the Wang et al dataset (why only the Wang dataset is analysed here also does not become clear). Is the relation still there if ER status is taken into consideration as a confounder.”

Unfortunately, and to the best of our knowledge, the Wang dataset is the only one among those used in this manuscript that allowed the information to be exported in a format compatible with GSEA analysis, and that is why it is the only one that we have used for this purpose.

As requested by the reviewer, we performed the analysis segregating ER+ and ER- tumors and the same trend was observed, although it only reached statistical significance in the case of the ER- samples. These new results appear in **Fig. R9**.

“13. Combining CXCR4 with FAAH to show prognostic value is misleading since FAAH in itself is already a marker providing the levels of significance observed here. It would be more fair to at least also show that CXCR4 on its own has significance since both markers are considered to have a shared relation to the measured end-point. Therefore combining them will not learn us anything. More informative would be to see how CXCR4 levels behave in the tumors that show a difference between the primary tumor and the lymph node metastasis for FAAH and those that do not.”

We agree with the reviewer that the survival curves showing the prognostic value of FAAH and CXCR4 in combination (former Fig. 6d and former Supplementary Fig. 9b) did not add much to their value as separate prognostic factors, and therefore they have been removed from the figure.

Regarding the correlation between *FAAH* and *CXCR4*, we have replaced in Fig. 6c the panel corresponding to TMA#2 for that previously shown in the former Supplementary Fig. 9c, which includes transcriptomic data from > 2500 patients and unequivocally shows the inverse association between these two genes.

“14. Unfortunately it do not become clear how FAAH is directly connected to CXCR4. Or at least it does not become clear of the effect of metAEA indeed phenocopied the effect of FAAH via the same pathway. Transcriptome or marker analyses should be done to show the same pathway are used. I am also concerned about the involvement of bonified cannabis receptor as gene expression of CNR and CNR2 is undetectable in BC tumors and cell lines (including the one used here). So I would appreciate the effect seen here is mediated by one of the cannabis receptor directly (e.g. by using knock-outs).”

We frankly agree with the reviewer that the weakest part of the manuscript was the mechanistic link between FAAH, AEA, CBRs and CXCR4. To address this key issue, we have performed additional *in vitro* and *in vivo* experiments with metAEA and pharmacological tools aimed to block those receptors:

- Previously, we had demonstrated that knocking-out FAAH in T-47D cells induced an EMT phenotype *in vitro* (Supplementary Fig. 7c), and that FAAH overexpression in MDA-MB-231 cells attenuated it (Supplementary Fig. 7d). In the **(new) Supplementary Fig. 10a**, we now demonstrate that metAEA treatment over a long period (4 weeks) also induces an EMT-like phenotype in T-47D cells through the activation of CBRs.
- Previously, we had demonstrated that lung-tropic MDA-MB-231 cells overexpressing FAAH formed less lung metastasis than parental cells (Fig. 4i). To demonstrate that this phenotype was due to lower endogenous AEA levels, we performed a first *in vivo* experiment in which mice were injected with FAAH-overexpressing lung-tropic MDA-MB-231 cells and subsequently treated with different doses of metAEA (0.1, 1 and 10 mg/kg). **Fig. R3** shows that metAEA, as parental cells with low levels of FAAH, increased the metastatic burden of these animals. Once the pro-metastatic effect of metAEA was demonstrated, we performed a second *in vivo* experiment to try to support the involvement of cannabinoid receptors and CXCR4 in metAEA action. The results of this experiment have been included in the revised version of the manuscript as **(new) Fig. 7d**, and show that the pro-metastatic effect of metAEA is prevented by CB₁R, CB₂R and CXCR4 selective antagonists, further reinforcing the participation of CBRs and CXCR4 in FAAH-driven metAEA-mediated actions.
- Previously, we had demonstrated that metAEA increased the invasive capacity of T-47D and MDA-MB-231 cells (Fig. 7b), and that this effect was prevented by blocking CB₁R and CB₂R with selective antagonists. We have now performed these experiments in MDA-MB-231 cells and obtained similar results, which are included in the revised manuscript as **Fig. 7c**.

Regarding the expression of CBRs in BC, Caffarel and colleagues provided the first direct evidence for the expression of CBRs in human breast tumors, with CB₁R and CB₂R mRNAs downregulated and upregulated, respectively, compared to noncancerous breast tissue (Caffarel *et al.*, 2006). Shortly after, Qamri and

colleagues detected CB₁R and CB₂R protein in human breast tumors as well as in the BC cell line MDA-MB-231 (Qamri *et al.*, 2009). To further confirm the expression of CBRs in the cell lines used in this study, Western blot assays have been performed and are shown in **Fig. R10**.

In addition to these results and those from the *in vitro* and *in vivo* experiments described in this point, the functional involvement of cannabinoid receptors in FAAH/metAEA-mediated actions is further supported by Figs. 7a, 7b, 7c, 7d and Supplementary Fig. 10.

References:

- Arimidex, T. (2008). *Effect of anastrozole and tamoxifen as adjuvant treatment for early-stage breast cancer: 100-month analysis of the ATAC trial. The lancet oncology*, 9(1), 45-53.
- Caffarel, M. M., Sarrió, D., Palacios, J., Guzmán, M., & Sánchez, C. (2006). Δ^9 -tetrahydrocannabinol inhibits cell cycle progression in human breast cancer cells through Cdc2 regulation. *Cancer research*, 66(13), 6615-6621.
- Chanrion, M., Negre, V., Fontaine, H., Salvetat, N., Bibeau, F., Grogan, G. M., ... & Darbon, J. M. (2008). *A gene expression signature that can predict the recurrence of tamoxifen-treated primary breast cancer. Clinical Cancer Research*, 14(6), 1744-1752.
- Chen, W., Hoffmann, A. D., Liu, H., & Liu, X. (2018). *Organotropism: new insights into molecular mechanisms of breast cancer metastasis. NPJ precision oncology*, 2(1), 4.
- Cravatt, B. F., Demarest, K., Patricelli, M. P., Bracey, M. H., Giang, D. K., Martin, B. R., & Lichtman, A. H. (2001). *Supersensitivity to anandamide and enhanced endogenous cannabinoid signaling in mice lacking fatty acid amide hydrolase. Proceedings of the National Academy of Sciences*, 98(16), 9371-9376.
- Early Breast Cancer Trialists' Collaborative Group. (1998). *Tamoxifen for early breast cancer: an overview of the randomised trials. The Lancet*, 351(9114), 1451-1467.
- Herschkowitz, J. I., Simin, K., Weigman, V. J., Mikaelian, I., Usary, J., Hu, Z., ... & Perou, C. M. (2007). *Identification of conserved gene expression features between murine mammary carcinoma models and human breast tumors. Genome biology*, 8, 1-17.
- Kaupila, S., Stenbäck, F., Risteli, J., Jukkola, A., & Risteli, L. (1998). *Aberrant type I and type III collagen gene expression in human breast cancer in vivo. The Journal of Pathology: A Journal of the Pathological Society of Great Britain and Ireland*, 186(3), 262-268.
- Kennecke, H., Yerushalmi, R., Woods, R., Cheang, M. C. U., Voduc, D., Speers, C. H., ... & Gelmon, K. (2010). *Metastatic behavior of breast cancer subtypes. Journal of clinical oncology*, 28(20), 3271-3277.
- Lim, E., Wu, D., Pal, B., Bouras, T., Asselin-Labat, M. L., Vaillant, F., ... & Visvader, J. E. (2010). *Transcriptome analyses of mouse and human mammary cell subpopulations reveal multiple conserved genes and pathways. Breast Cancer Research*, 12(2), 1-14.
- Lopez-Knowles, E., Pearson, A., Schuster, G., Gellert, P., Ribas, R., Yeo, B., ... & Dowsett, M. (2019). *Molecular characterisation of aromatase inhibitor-resistant advanced breast cancer: the phenotypic effect of ESR1 mutations. British journal of cancer*, 120(2), 247-255.
- Lower, E. E., Glass, E. L., Bradley, D. A., Blau, R., & Heffelfinger, S. (2005). *Impact of metastatic estrogen receptor and progesterone receptor status on survival. Breast cancer research and treatment*, 90, 65-70.

Martin, L. A., Ribas, R., Simigdala, N., Schuster, E., Pancholi, S., Tenev, T., ... & Dowsett, M. (2017). Discovery of naturally occurring ESR1 mutations in breast cancer cell lines modelling endocrine resistance. *Nature communications*, 8(1), 1865.

Ogba, N., Manning, N. G., Bliesner, B. S., Ambler, S. K., Haughian, J. M., Pinto, M. P., ... & Horwitz, K. B. (2014). Luminal breast cancer metastases and tumor arousal from dormancy are promoted by direct actions of estradiol and progesterone on the malignant cells. *Breast Cancer Research*, 16, 1-14.

Ono, M., Sakamoto, M., Ino, Y., Moriya, Y., Sugihara, K., Muto, T., & Hirohashi, S. (1996). Cancer cell morphology at the invasive front and expression of cell adhesion-related carbohydrate in the primary lesion of patients with colorectal carcinoma with liver metastasis. *Cancer: Interdisciplinary International Journal of the American Cancer Society*, 78(6), 1179-1186.

Pfefferle, A. D., Herschkowitz, J. I., Usary, J., Harrell, J. C., Spike, B. T., Adams, J. R., ... & Perou, C. M. (2013). Transcriptomic classification of genetically engineered mouse models of breast cancer identifies human subtype counterparts. *Genome biology*, 14, 1-16.

Rochefort, H., Platet, N., Hayashido, Y., Derocq, D., Lucas, A., Cunat, S., & Garcia, M. (1998). Estrogen receptor mediated inhibition of cancer cell invasion and motility: an overview. *The Journal of steroid biochemistry and molecular biology*, 65(1-6), 163-168.

Qamri, Z., Preet, A., Nasser, M. W., Bass, C. E., Leone, G., Barsky, S. H., & Ganju, R. K. (2009). Synthetic cannabinoid receptor agonists inhibit tumor growth and metastasis of breast cancer. *Breast Tumor Suppressive Effects of Cannabinoids. Molecular cancer therapeutics*, 8(11), 3117-3129.

Szostakowska, M., Trębińska-Stryjewska, A., Grzybowska, E. A., & Fabisiewicz, A. (2019). Resistance to endocrine therapy in breast cancer: molecular mechanisms and future goals. *Breast Cancer Research and Treatment*, 173, 489-497.

Tabor, S., Szostakowska-Rodzios, M., Fabisiewicz, A., & Grzybowska, E. A. (2020). How to predict metastasis in luminal breast cancer? Current solutions and future prospects. *International Journal of Molecular Sciences*, 21(21), 8415.

Figure R1

Figure R1. Expression of *FAAH* and *ESR1* positively correlate in human BC. **a-c** Scatter plots (a, b) and heat map (c) showing the positive correlation between *FAAH* and *ESR1* mRNA expression in human BC samples according to the TCGA (Cancer Genome Atlas, 2012) (a), METABRIC (Curtis *et al.*, 2012) (b) and bc-GenExMiner (Jezequel *et al.*, 2021) (c) datasets. **d** Scatter plot showing the positive correlation between *FAAH* and *ESR1* mRNA expression in human BC cell lines according to the dataset published by Neve *et al.*, 2006.

Figure R2

Figure R2. Low FAAH expression in luminal breast tumors is associated with poor patient prognosis. Kaplan-Meier curves for overall survival in luminal BC samples with high and low FAAH expression obtained from TMA #1.

Figure R3

Figure R3. metAEA treatment induces pro-metastatic responses in athymic mice injected with lung-tropic MDA-MB-231 cells. Bar graph representing lung histology of athymic female mice after injection of FAAH-modulated MDA-MB-231 cells in the tail vein and subsequently treated with increasing doses of metAEA. Scores measure the percentage of metastatic penetrance, which is the % of lung parenchyma invaded by cancer cells.

Figure R4

Figure R4. Low FAAH expression in luminal breast tumors is associated with higher probability of relapse. a Kaplan-Meier curves for relapse-free survival in luminal BC samples with high and low FAAH expression obtained from the dataset published in Chanrion *et al.*, 2008. **b** Relative mRNA expression of FAAH in luminal BC samples of patients that suffered or did not suffer relapse according to the dataset published in Chanrion *et al.*, 2008.

Figure R5A

Figure R5A. Magnification of the FAAH staining in the Fig. 1c.

Primary tumor

Lymph node

Patient 42

Patient 124

Figure R5B. Magnification of the FAAH staining in the Fig. 2e.

Figure R6

Figure R6. Low FAAH expression at the invasion front is associated with ER loss. On the left, FAAH staining of patient 42 as represented in Supplementary Fig. 5d. On the right, ER staining of the same patient. Arrows indicate cells with lower FAAH and ER expression at the invasion front. Scale bar = 250 μ m.

Figure R7A

Figure R7A. The negative correlation between mRNA expression of *FAAH* and some of the BC-related genes included the PCR Array of Mouse Breast Cancer is also found in clinical samples. Scatter plots showing the negative correlation between mRNA expression of *FAAH* and some of the BC-related genes included the PCR Array of Mouse Breast Cancer in human BC samples according to the METABRIC dataset (Curtis *et al.*, 2012).

Figure R7B

Figure R7B. The negative correlation between mRNA expression of *FAAH* and some of the metastasis-related genes included the PCR Array of Human Tumor Metastasis is also found in clinical samples. Scatter plots showing the negative correlation between mRNA expression of *FAAH* and some of the metastasis-related genes included the PCR Array of Human Tumor Metastasis in human BC samples according to the METABRIC dataset (Curtis *et al.*, 2012).

Figure R7C

Figure R7C. The negative correlation between mRNA expression of *FAAH* and some of the DEGs from the RNA-seq of *FAAH*-modulated T-47D cells is also found in clinical samples. Scatter plots showing the negative correlation between mRNA expression of *FAAH* and some of the differentially expressed genes (DEGs) in the RNA-seq of *FAAH*-modulated T-47D cells according to the METABRIC dataset (Curtis *et al.*, 2012).

Figure R8

Figure R8. Expression of *FAAH* and *CXCR4* negatively correlate in both ER+ and ER- BC samples. Scatter plots showing the correlation between mRNA expression of *ESR1* and *CXCR4* (left), *ESR1* and *FAAH* (middle) and *FAAH* and *CXCR4* (right) in human ER+ and ER- BC samples according to the METABRIC dataset (Curtis *et al.*, 2012). Correlation between *FAAH* and *CXCR4* expression has been highlighted in red.

Figure R9

CXCR4 pathway gene set
(Schaefer *et al.*, 2009)

Figure R9. GSEA showing an activation of the CXCR4 pathway established in Schaefer *et al.*, 2009, in low FAAH-expressing ER+ and ER- samples according to the dataset published in Wang *et al.*, 2005.

Figure R10

Figure R10. Expression of cannabinoid receptors in BC cell lines. WB analysis of CB₁R and CB₂R in a panel of human BC cell lines representative of the main intrinsic subtypes. Densitometric values after normalization against α-Tubulin and control condition (MCF7 cell line) are depicted in purple.

MULTIVARIATE TEST FOR TMA #1

Cox regression

Case Processing Summary

		N	Percent
Cases available in analysis	Event ^a	22	5,9%
	Censored	263	70,3%
	Total	285	76,2%
Cases dropped	Cases with missing values	77	20,6%
	Cases with negative time	0	0,0%
	Censored cases before the earliest event in a stratum	12	3,2%
	Total	89	23,8%
Total		374	100,0%

a. Dependent Variable: Follow

Categorical Variable Codings^{a,c,d,e,f}

		Frequency	(1)
ER ^b	0=ER-	59	0
	1=ER+	238	1
PR ^b	0=PR-	98	0
	1=PR+	199	1
HER2 ^b	0=HER2-	188	0
	1=HER2+	109	1
FAAH ^b	0=FAAH LOW	204	0
	1=FAAH HIGH	93	1
TN ^b	0=TN-	273	0
	1=TN+	24	1

a. Category variable: ER

b. Indicator Parameter Coding

c. Category variable: PR

d. Category variable: HER2

e. Category variable: FAAH

f. Category variable: TN

Block 0: Starting block.

Omnibus Tests of Model Coefficients

-2 Log Likelihood
215,342

Block 1: Method = Enter

Omnibus Tests of Model Coefficients^a

-2 Log Likelihood	Overall (score)			Change From Previous Step			Change ... Chi-square
	Chi-square	df	Sig.	Chi-square	df	Sig.	
184,137	31,616	5	<,001	31,206	5	<,001	31,206

Omnibus Tests of Model Coefficients^a

Change From Previous ...

df	Sig.
5	<,001

a. Beginning Block Number 1. Method = Enter

Variables in the Equation

	B	SE	Wald	df	Sig.	Exp(B)	95,0% CI for Exp(B)	
							Lower	Upper
FAAH	-1,728	,779	4,915	1	,027	,178	,039	,819
ER	,113	,585	,037	1	,847	1,119	,356	3,522
TN	,703	,873	,648	1	,421	2,019	,365	11,174
PR	-2,098	,610	11,843	1	<,001	,123	,037	,405
HER2	,482	,597	,651	1	,420	1,619	,502	5,220

Covariate Means

	Mean
FAAH	,316
ER	,796
TN	,081
PR	,670
HER2	,375

REVIEWERS' COMMENTS

Reviewer #1 (Remarks to the Author):

No concerns

Reviewer #3 (Remarks to the Author):

No further comments

RESPONSE TO REVIEWERS' COMMENTS

POINT-BY-POINT ANSWER TO REVIEWER #1 (Remarks to the Author)

No concerns

We thank the reviewer for his/her positive comments on our manuscript.

POINT-BY-POINT ANSWER TO REVIEWER #3 (Remarks to the Author)

No further comments

We thank the reviewer for his/her positive comments on our manuscript.